# Regime Shifts in Arctic Terrestrial Hydrology Manifested From Impacts of Climate Warming

Michael A. Rawlins[1] and Ambarish V. Karmalkar[2,1]

[1]Department of Earth, Geographic, and Climate Sciences, University of Massachusetts, Amherst, MA 01003, USA
[2]Department of Geosciences, University of Rhode Island, Kingston, RI 02881, USA

*Correspondence to:* Michael A. Rawlins (mrawlins@umass.edu)

## Abstract

Anthropogenic warming in the Arctic is causing hydrological cycle intensification and permafrost thaw, with implications for flows of water, carbon, and energy from terrestrial biomes to coastal zones. To better understand likely impacts of the changes, we used a hydrology model driven by meteorological data from atmospheric reanalysis and two global climate models for the period 1980–2100. The hydrology model accounts for soil freeze-thaw processes and was applied across the pan-Arctic drainage basin. The simulations point to greater changes over northernmost areas of the basin underlain by permafrost, and the western Arctic. An acceleration of simulated river discharge over the recent past is commensurate with trends drawn from observations and reported in other studies. Between early (2000–2019) and late century (2080–2099) the model simulations indicate an increase in annual total runoff of 17–25%, while the proportion of runoff emanating from subsurface pathways is projected to increase 13–30%, with the largest changes noted in summer and autumn, and across areas with permafrost. Most notably, runoff contributions to river discharge shift to northern parts of the Arctic basin that contain greater amounts of soil carbon. Each season sees an increase in subsurface runoff, spring is the only season where surface runoff dominates the rise in total runoff, and summer experiences a decline in total runoff, despite

an increase in the subsurface component. The greater changes that are seen in areas where permafrost exists supports the notion that increased soil thaw is shifting hydrological contributions to more subsurface flow. The manifestations of warming, hydrological cycle intensification, and permafrost thaw will impact Arctic terrestrial and coastal environments through altered river flows and the materials they transport.

# 1 Introduction

Hydrological cycle intensification and permafrost thaw are among a myriad of environmental changes reshaping the Arctic environment (Rawlins et al., 2010; Hinzman et al., 2013; Box et al., 2019; Overland et al., 2019). Climate forcings including increasing air temperature and precipitation are key drivers of alterations to the Arctic system (Box et al., 2019). The Arctic has warmed 2.5 to 4 times faster than the global average over the past several decades (Rantanen et al., 2022; Wang et al., 2022) and experienced substantial decreases in Arctic Ocean sea ice extent and volume (Stroeve and Notz, 2018; Serreze and Meier, 2019). Warming is leading to hydrologic intensification that is projected to drive higher precipitation rates (Bintanja and Selten, 2014; McCrystall et al., 2021), with concomitant rises in river discharge (Shiklomanov and Shiklomanov, 2003; Dankers and Middelkoop, 2008). Permafrost thaw has the potential to change how water is stored and moved, and to mobilize vast stores of organic carbon sequestered in soils (Frey and Smith, 2005; Koch et al., 2022; Mohammed et al., 2022; Del Vecchio et al., 2024), and rising river discharge (Peterson et al., 2002; Wagner et al., 2011; Feng et al., 2021) furthermore imply associated changes in exports of water, energy, carbon, and other constituents to coastal zones (Tank et al., 2016; Behnke et al., 2021; Zhang et al., 2021). In light of these alterations, it is important to better understand how climate warming, hydrological cycle intensification, and permafrost thaw will impact Arctic terrestrial hydrology and, in turn, exports of freshwater and associated materials through the Arctic drainage basin and into coastal zones.

The seasonal storage of precipitation in the form of snow is a defining element of Arctic hydrology, contributing to abundant surface water storages and high river flows following spring melt. The presence of permafrost is another important element influencing the region's water cycle. Climate warming is intensifying Earth's water cycle, increasing precipitation, evaporation, evapotranspiration (ET), and river discharge globally (Huntington, 2006, 2010), and across Arctic regions (Peterson et al., 2002; Déry et al., 2009; Rawlins et al., 2010). Intensification or "acceleration" involves the effects of both atmospheric moisture holding capacity and moisture avail-

ability. Declining sea ice is making the Arctic Ocean and its surrounding seas a more available source of moisture, with locally-driven precipitation recycling greatest in winter across the Beaufort-Chukchi, Laptev, Kara, and East Siberian Seas (Ford and Frauenfeld, 2022). Increasing late summer precipitation and a shift toward rainfall runoff is occurring across watersheds in northwest Alaska (Arp et al., 2020; Rawlins, 2021; Arp and Whitman, 2022). Terrestrial hydrology in the Arctic is also strongly controlled by the presence of permafrost and the seasonal thawing and freezing of soils (Tananaev et al., 2020). Permafrost underlies approximately one fifth of the global land area and influences processes involving runoff, aquatic biogeochemistry (Frey and McClelland, 2009; Spencer et al., 2015; Hu et al., 2023), and land-atmosphere greenhouse gas exchanges (Christensen et al., 2004; McKenzie et al., 2021). Permafrost acts as an impermeable hydrological barrier, helping to maintain high soil suprapermafrost moisture levels while reducing soil water storage capacity and constraining subsurface flow (Woo et al., 2008; Walvoord and Kurylyk, 2016). The presence or absence of permafrost and variability in precipitation processes lead to varying amounts of surface and subsurface runoff contributions to river discharge and, in turn, land-ocean exports of freshwater and associated materials. Warming is causing long-term changes in near-surface soil freeze/thaw cycles and permafrost (Anisimov and Reneva, 2006; Koven et al., 2013; Guo et al., 2018; Peng et al., 2018; Biskaborn et al., 2019), with implications for permafrost hydrology (Woo et al., 2008; Liljedahl et al., 2016; Lafrenière and Lamoureux, 2019; Jin et al., 2022). Subsidence due to thawing soils will likely lead to more runoff, while significantly accelerating drying of tundra landscapes in a warming climate (Painter et al., 2023). Studies suggest that permafrost degradation leads to increased moisture transport from the surface to deeper soils, potentially contributing to increased river baseflows (Walvoord and Striegl, 2007) and cold season discharge (St. Jacques and Sauchyn, 2009; Shiklomanov et al., 2013; Tananaev et al., 2016; Rawlins et al., 2019; Debolskiy et al., 2021; Wang et al., 2021; Liu et al., 2022). In northwest Alaska, positive trends in air temperature and precipitation are greatest in autumn which, together with permafrost thaw, is likely leading to enhanced subsurface "suprapermafrost" runoff during that time (Rawlins, 2021).

Climate models are essential tools for understanding how manifestations of climate warming will alter the Arctic's terrestrial hydrology and riverine land-ocean fluxes. Model projections point to future precipitation increases over the 21$^{st}$ century through enhanced regional evaporation and poleward moisture transport (Bintanja et al., 2020), and sea ice declines (Bintanja and Selten, 2014). Models with the strongest warming response point to decreased snowfall across the high (70–90 °N) Arctic. The precipitation increases are firmly linked to Arctic warming and sea-ice

decline (Bintanja, 2018; Arp et al., 2020), and are likely to increase river discharge (Peterson et al., 2002; Zhang et al., 2013). Recent coordinated research programs have produced bias-corrected climate model data for historical and future conditions from consistent protocol frameworks (Warszawski et al., 2014; Lange, 2021). Simulations of permafrost dynamics and associated soil freeze-thaw processes require attention to several key processes absent in many land-surface models (Alexeev et al., 2007; Nicolsky et al., 2007; Lawrence and Slater, 2008). Slater and Lawrence (2013) concluded that, in general, permafrost is not well represented by the ensemble of CMIP5 models. Examining permafrost dynamics in global models participating in the CMIP6, Burke et al. (2020) found that simulation of active-layer thickness (ALT) and other key features often fell outside the observed range, with errors attributable to shallow and poorly resolved soil profiles and structural weaknesses in snow physics and soil hydrology within some of the models.

In this study we use simulations with a permafrost hydrology model with sophisticated soil-freeze thaw algorithms that represent an improvement upon traditional land-surface models to evaluate how climate alterations linked to warming, primarily hydrological cycle intensification and permafrost thaw, will influence Arctic terrestrial hydrology and, in turn, land-ocean riverine freshwater and biogeochemical fluxes. We begin by examining meteorological data from climate models to understand the atmospheric forcings and their influence on surface hydrology. Model simulations are validated against select observations for sublimation, ET, ALT, and river discharge. We then examine changes over the 21$^{st}$ century to gain insights into how hydrological cycle intensification and permafrost thaw will impact key elements of Arctic terrestrial hydrology controlling river exports, and test the hypothesis that within the Arctic drainage basin, changes in subsurface runoff are greatest in permafrost areas.

# 2 Methods

## 2.1 Study area and spatial grid

The pan-Arctic drainage basin used in this study encompasses approximately 22.45 million square kilometers. It has a wide range of land cover types, from grasslands in southern Canada and central Eurasia to boreal forests to tundra in far northern areas. This domain includes basins of rivers draining to the Arctic Ocean, Hudson Bay, and the Bering Strait, with the large Yukon River draining into the latter. The region's four largest rivers—the Ob, Yenesey, Lena, and Mackenzie—flow primarily in a south-to-north direction, and account for roughly half (49%) of the

pan-Arctic basin area. Model forcing data, simulations, and outputs were produced on the 25×25 km EASE-Grid (Brodzik and Knowles, 2002). The spatial domain encompassing the terrestrial pan-Arctic as defined in this study has 35,693 grid cells. Each grid cell has 23 vertical layers extending to 60 m depth in which water and energy interact with the soil and vegetation. Thus the model is set up and executed in three dimensions (2 D horizontal and 1 D vertical) like many similar land surface models often used to quantify terrestrial hydrological fluxes.

## 2.2   Modeling approach

The modeling approach leverages simulations with the Permafrost Water Balance Model (PWBM v4) to investigate the impacts of warming, hydrological cycle intensification, and permafrost thaw on terrestrial hydrological fluxes within and through the pan-Arctic drainage basin. Many of the details of PWBM have been documented elsewhere, so a general description is provided here with the reader encouraged to obtain more detail from the cited literature. The PWBM simulates all major elements of the water cycle, including transpiration and soil and surface-water evaporation, snow storage, sublimation (Rawlins et al., 2003, 2013), runoff (Rawlins et al., 2021), and soil freeze-thaw. Past applications include assessment of causes behind record Eurasian discharge (Rawlins et al., 2010); estimation of surface water dynamics (Schroeder et al., 2010); analysis of present and future water budgets (Clilverd et al., 2011); quantification of freshwater and dissolved organic carbon fluxes (Rawlins et al., 2021); investigation of trends in those fluxes to a coastal lagoon in northwest Alaska (Rawlins, 2021); and exploration of the links between surface organic soil properties and moisture dynamics across the Alaska North Slope (Yi et al., 2022). PWBM operates at an implicit daily time step, with meteorological forcings (air temperature, precipitation, wind speed) typically drawn from reanalysis data for regional-scale simulations or, when applied to smaller watersheds, meteorological station data. Daily simulated ET depends on atmospheric demand and surface and soil conditions. In this study we applied the Hamon function to estimate potential evapotranspiration. The model includes a surface water pool that is typically transient and most often occurs after snowmelt. Runoff is generated when (i) the amount of available water at the surface exceeds infiltration capacity and (ii) the amount of water in a soil layer exceeds field capacity, which is a function of soil texture. The sum of surface and subsurface runoff from one or more soil layers within a grid cell constitutes daily total runoff. We use the term "subsurface runoff" for the water flux that has followed subsurface pathways into the stream. Subgrid fraction of inundated area (lakes and ponds) are parameterized based on observed

data (Du et al., 2016), with total runoff across each grid cell calculated as a weighted total from the inundated and non-inundated areas. We applied a simple river flow accumulation and linear routing model (Rawlins et al., 2019) to estimate the timing shift in discharge export at the coast. The snow model simulates the effects of seasonal changes in snow density and, in turn, snow thermal conductivity (Liston et al., 2007; Sturm et al., 1995). Soil freeze-thaw process representations include a multi-layer soil module with algorithms for unfrozen water dynamics and phase change, as well as specification of the thermal and hydrological properties of organic soils (Sazonova and Romanovsky, 2003; Nicolsky et al., 2007). The PWBM has a 60 meter soil column, includes parameterizations for thermal and hydraulic properties of organic soils, and simulates the effect of depth hoar and wind compaction on snow density. Rawlins et al. (2013) describe the soil freeze-thaw and snow algorithms, and calibration procedures, which involve factors controlling ET, snow sublimation, and subsurface runoff that differ between forest and tundra landscapes. In this study each transient simulation was preceded by a 50-year spinup on year 1980 to stabilize soil temperature, moisture, and soil dissolved organic carbon (DOC) pools. While parameterizations for fields such as soil texture and vegetation cover are fundamental elements of land surface and hydrological model simulations, simulated runoff in Arctic regions is most sensitive to the time-varying meteorological forcings (Rawlins et al., 2003).

Permafrost extent is based on end of season soil temperatures. If the soil column down to the maximum 60 meter depth is frozen, beneath a thawed upper zone (i.e. active layer), the grid cell is deemed to have permafrost that year. Thus permafrost state is a binary classification. In the case where soil temperatures are well simulated, one can assume that there is discontinuous permafrost in regions where many grid cells classified as permafrost interface with many grid cells classified as seasonally frozen. The impact of subsidence on permafrost thaw is not accounted for in the simulations, though the effect may be relatively small (Painter et al., 2023), particularly in areas lacking polygonal tundra. In models operating at continental scales, estimates of permafrost extent across transition zones between continuous permafrost and the non-permafrost areas are more uncertain due to limitations resolving spatial variations.

## 2.3   Meteorological forcings

This study focuses on numerical model simulations that were forced with gridded meteorological data (Table 1). We begin by examining simulations forced with reanalysis data to characterize dynamics over the recent past. Changes over the 21$^{st}$

century were assessed using simulations forced with meteorological data from coupled climate models, rather than the hydrology (eg. runoff) from them, as outputs from individual models can vary widely, and often imply unrealistic long-term systematic changes in water storage and level within entire basins (Bring et al., 2015).

Table 1: Simulations conducted in the study, time period for the transient simulation, and origin of forcing data. Each transient simulation was preceded by a 50 year spinup. For the climate model forcing, the 1980–2100 period includes two different experiments.

| Model simulations | | |
|---|---|---|
| **Name** | **Period** | **Forcing** |
| PWBM-W5E5 | 1980–2019 | Bias-adjusted ECMWF Reanalysis v5 (ERA5) |
| PWBM-ERA5 | 1980–2019 | ERA5 Reanalysis |
| PWBM-MERRA | 1980–2013 | Modern-Era Retrospective Analysis for Research and Applications |
| PWBM-IPSL | 1980–2100 | IPSL-CM6A-LR (Historical: 1980–2014, SSP3-7.0: 2015–2100) |
| PWBM-MPI | 1980–2100 | MPI-ESM1-2-HR (Historical: 1980–2014, SSP3-7.0: 2015–2100) |

Simulations were made using forcings from three reanalysis datasets (W5E5, ERA5, MERRA) and two global climate models from the Coupled Model Intercomparison Project Phase 6 (CMIP6). The WFDE5 data—WATCH Forcing Data methodology applied to ERA5 reanalysis—is bias-adjusted ERA5 data at $0.5° \times 0.5°$ spatial and sub-daily resolutions, generated specifically to be used as climate data inputs for impacts studies (Cucchi et al., 2020). The WFDE5 over land is merged with ERA5 over the ocean to produce W5E5 data (Lange, 2019), compiled as part of phase 3b of the Inter-Sectoral Impact Model Intercomparison Project (ISIMIP3b) (Lange, 2019, 2021). We downloaded and analyzed W5E5 version 2 data for use as meteorological forcings for simulations over the historical period. We use bias-adjusted data (W5E5 v2 and climate models) prepared as part of the ISIMIP framework (Cucchi et al., 2020; Lange et al., 2021). We also applied data from ERA5 and MERRA reanalysis to gauge the accuracy of the air temperature (2 m), precipitation, and wind speed forcings and for model validation. Precipitation amounts in the W5E5 data are lowest among the three reanalysis datasets. To ameliorate this bias in the simulation forced with W5E5 we increased each precipitation value by 20%. The ISIMIP3b climate model forcing data are bias adjusted and statistically downscaled, and available for five CMIP6 models (GFDL-ESM4, IPSL-CM6A-LR, MPI-ESM1-2-HR, MRI-ESM2-0, UKESM1-0-LL) forced with three Shared Socioeconomic Pathways (SSP) scenarios (SSP1-2.6, SSP3-7.0, SSP5-8.5). In our two simulations over years 1980–2100 we used data from two models (MPI-ESM1-2-HR, IPSL-CM6A-LR) forced with SSP3-7.0, which is a high emissions "business as usual" scenario, and suitable to investigate the response of Arctic hydrology to a strong climate forcing. These two climate models generally bracket the range of climate projections for the pan-Arctic region across the five CMIP6 models (Fig. S1). The selection of these two models—hereafter IPSL and MPI—is aimed at capturing a wide range of temperature and precipitation projections, but not necessarily the full range. Air temperature and precipitation changes expressed by the models are described in Sect. 4.1 and 4.2 respectively. In a study examining which CMIP3 models performed best at capturing meteorological quantities across parts of the Arctic, a predecessor of the MPI-ESM ranked highest (Walsh et al., 2008).

## 2.4 Statistical analysis

Our analysis of changes closely connected to Arctic rivers centers on differences between 20-yr intervals representing early (2000–2019) and late (2080–2099) century conditions. Specifically we mapped climatological averages over these periods and examined the differences for each domain grid cell. Domain-wide averages were com-

puted from all 35,693 grid cells covering the domain. The statistical significance of differences between the two periods were calculated for select quantities. Before applying the statistical significance test we used graphical analysis and the Shapiro–Wilk test (Shapiro and Wilk, 1965) to determine if the data series of interest is approximately normally distributed. The paired t test was then applied to test the null hypothesis that the mean difference between two variables is zero. Relative (percentage) difference is calculated based on the standard formula: Relative difference $(\%) = (Z_2 - Z_1) / Z_1$ x 100, where $Z_1$ and $Z_2$ are values for early and late periods respectively.

Metrics which rely on squared differences are known to be problematic (Willmott and Matsuura, 2005; Hodson, 2022). The RMSE in particular is inappropriate because it is a function of three characteristics of a set of errors, rather than of one (the average error). RMSE varies with the variability within the distribution of error magnitudes and with the square root of the number of errors, as well as with the average-error magnitude (MAE). Interpretation problems can thus arise because sums-of-squares-based statistics do not satisfy the triangle inequality (Willmott and Matsuura, 2009). Thus MAE and mean bias error (MBE) are more natural measures of average error, and evaluations and inter-comparisons in this study are based upon it.

In this study we leverage the simulations forced by the two climate models to investigate the sensitivity of thermal and hydrological responses to different climate forcings, not to provide robust quantitative projections, which would require a multi-model, multi-scenario ensemble.

# 3 Model Validation

We first compared key components of the simulated water budget–active-layer thickness, sublimation, evapotranspiration, and discharge–with different observational datasets to assess the credibility of the PWBM simulations. Simulated active-layer thickness (ALT) and model-estimated permafrost extent is compared to ALT data from the National Tibetan Plateau/Third Pole Environment Data Center (TPDC) (Fig. 1a–d) and permafrost area from International Association of Permafrost (IPA) data. In this study the active layer is the top layer of ground subject to annual thawing and freezing in areas underlain by permafrost. Simulated ALT in the model simulations spans a greater range compared with the TPDC data (Fig. 1e). However, the TPDC ALT estimates are known to have a reduced distribution range owing to the machine learning approach used (Ni et al., 2021). As Ran et al. (2022) described in their analysis of the TPDC dataset, the uncertainty of ALT is considerable, espe-

cially in the vast area of western Siberia where the training data are sparse. Further, they suggested that the low spatial representativeness of training data may have led to an overestimation in several Siberian mountain regions and underestimation near the lower boundary of permafrost. Moreover, in situ ALT is obtained at a point location that may not be representative of the region in which it is located. In light of these uncertainties, permafrost extent is generally well captured, with differences from total area of continuous and discontinuous permafrost in the IPA dataset of less than 10% (Table 2). For comparison, the fraction of continuous, discontinuous, and sporadic/isolated permafrost within the major river basins is shown in Table 3. In Eurasia there exists a clear west-east gradient, with the relatively cold Lena basin having a large amount of continuous permafrost. In North America the Mackenzie basin has a large extent of land in the south devoid of permafrost, a reflection of the relatively warm climate there.

We used the simulation forced with W5E5 data (PWBM-W5E5) to evaluate the magnitude of vertical fluxes of water from sublimation and ET over the recent past (Fig. 2). Overestimates in simulated sublimation (Figure S2a) are noted (domain-wide average sublimation of 40 mm $yr^{-1}$ for GLEAM and 57 mm $yr^{-1}$ for PWBM-W5E5), though the discrepancy is small relative to the magnitudes of annual total runoff and ET (MAE = 27 mm $yr^{-1}$). Simulated ET (260 mm $yr^{-1}$) generally falls between the estimates from GLEAM (304 mm $yr^{-1}$) and remote sensing-based data (230 mm $yr^{-1}$), differences of 14% and 12% (MAE of 64 and 198 mm $yr^{-1}$) respectively. The model generally captures the spatial pattern in sublimation and ET, though regionally there are notable differences, particularly across the warmer southerly areas where PWBM tends to underestimate ET (Figure S2b,c). For runoff this result points to a possible wet bias in those areas relative to observed conditions.

We compared simulated discharge volume to a new dataset, the Remotely-sensed Arctic Discharge Reanalysis (RADR), that was generated through assimilation of approximately 9.18 million discharge observations derived from 227 million river width measurements from Landsat images (Feng et al., 2021). Simulated discharge volume is the sum total of runoff over the contributing river basin. This evaluation was performed for total discharge from the pan-Arctic drainage basin and five large Arctic rivers: the Ob, Yenesey, Lena, Mackenzie, and Yukon (Fig. S3). The model tends to overestimate discharge across western Eurasia and underestimate it across eastern Eurasia. Differences are modest for the two North American rivers. Yet the magnitude of pan-Arctic discharge is well constrained. Average freshwater export to the Arctic Ocean from the study domain over the period 1984–2018 is 5,169 $km^3$ $yr^{-1}$ based on RADR. Over the same period, annual total discharge is 5752, 5822,

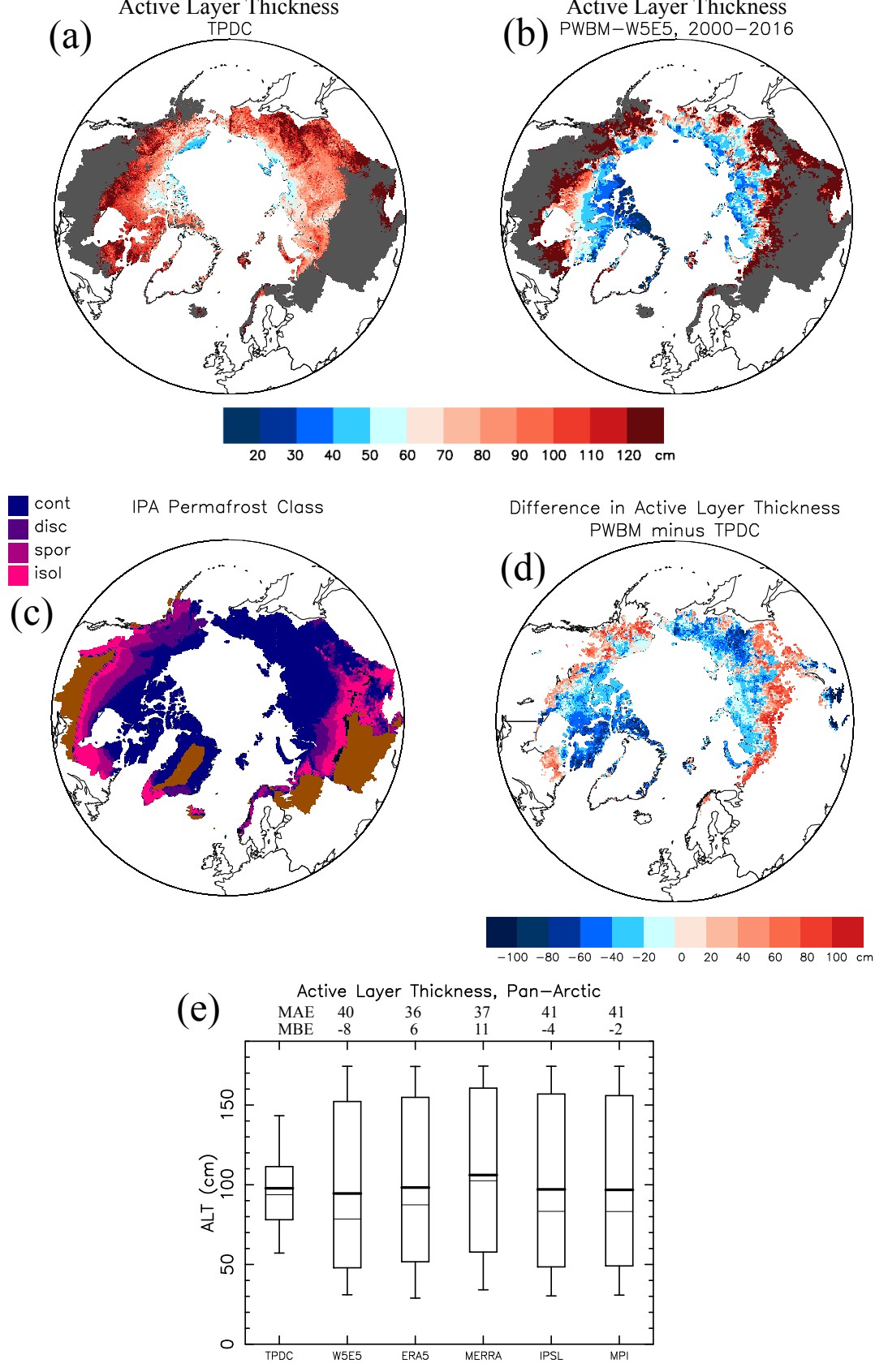

Figure 1: (a) Active-layer thickness (ALT, cm) from the TPDC database (Ran et al., 2022) for the period 2000–2016, and (b) from the PWBM simulation forced with W5E5 data over same period. Grey shading indicates non-permafrost areas. (c) Permafrost classification from International Association of Permafrost (IPA) data. (d) Difference in ALT (cm) between PWBM and TPDC. (e) Distributions of annual maximum ALT (cm) for all grids with permafrost. ALT is the average for each year over the period 2000–2016. TPDC is used as validation for the ALT estimated by simulations forced with data from W5E5, ERA5, MERRA (2000–2013), IPSL, and MPI. Boxplot rectangles bracket the 25th and 75th percentiles. Whiskers extend to the 5th and 95th percentiles. Thick and thin horizontal lines mark the distribution mean and median respectively. Mean absolute error (cm) and mean bias error (cm) shown.

Table 2: Permafrost areal extent and difference from observed extent across the study domain. Area in million km$^2$ from the International Permafrost Association (IPA) classification (Brown et al., 2001), the National Tibetan Plateau Data Center (TPDC) dataset (Ran et al., 2022), and PWBM simulations. Areas of continuous and discontinuous permafrost were added for the IPA estimate. Difference is defined based on observations from the IPA-based extent. For the simulated estimates, a grid cell is deemed to have permafrost under the standard definition of ground (model soil layer) that remains at or below 0°C for at least 2 consecutive years.

| Data | Area ($10^6$ km$^2$) | Difference (%) |
|------|------|------|
| IPA | 13.2 | – |
| TPDC | 12.5 | −5.5 |
| PWBM-W5E5 | 12.7 | −4.2 |
| PWBM-ERA5 | 13.1 | −0.8 |
| PWBM-MERRA | 10.5 | −20.4 |
| PWBM-IPSL | 12.4 | −6.2 |
| PWBM-MPI | 11.8 | −10.9 |

Table 3: Permafrost coverage by class in percent (%) for major river basins of the terrestrial Pan-Arctic. The fraction of land without permafrost is in column non-PF.

| Basin | continuous | discontinuous | sporadic/isolated | non-PF |
|------|------|------|------|------|
| Ob | 4.3 | 3.8 | 5.0 | 86.9 |
| Yenesei | 31.9 | 11.0 | 51.9 | 5.2 |
| Lena | 77.4 | 12.9 | 9.4 | 0.3 |
| Mackenzie | 15.7 | 29.6 | 47.3 | 7.4 |
| Yukon | 18.8 | 68.1 | 13.1 | 0.0 |

and 5811 km$^3$ yr$^{-1}$ in the simulations forced by W5E5, IPSL, and MPI respectively (Fig. S4), giving differences from RADR discharge of less than 13%. The simulation forced with W5E5 captures the acceleration in Arctic discharge reported in other studies (Peterson et al., 2002; Feng et al., 2021). The linear trend of 8.3 km$^3$ yr$^{-2}$ (0.15% yr$^{-1}$) closely aligns with the acceleration (11.6 km$^3$ yr$^{-2}$, 0.22% yr$^{-1}$) from RADR discharge (Feng et al., 2021), and is in the upper range of estimates (3.5–10 km$^3$ yr$^{-2}$) from prior studies (Shiklomanov et al., 2000; McClelland et al., 2006; Rawlins et al., 2010). For comparison, an analysis for the four largest Arctic-draining rivers (Mackenzie, Ob, Yenisei, and Lena) indicates that the combined annual discharge increased by 89 km$^3$ decade$^{-1}$ over the period 1980–2009, amounting to an approximate 14% increase over the 30-year period (Ahmed et al., 2020). Hydrological cycle intensification is connected with warming, and also manifested by increases

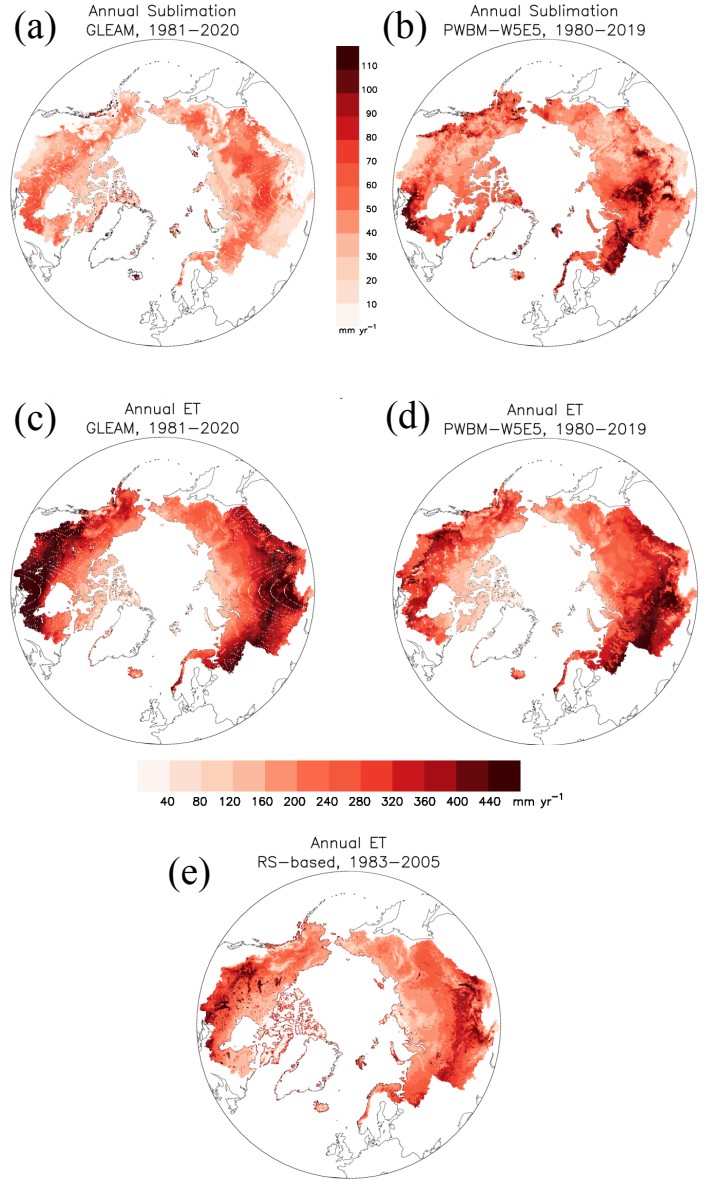

Figure 2: (a) Annual total sublimation (mm yr$^{-1}$) and (c) evapotranspiration (ET, mm yr$^{-1}$) from GLEAM (Miralles et al., 2011; Martens et al., 2017) and PWBM-W5E5 (b,d). Bottom panel (e) shows ET from a dataset derived from remote sensing data (Zhang et al., 2009).

in vertical fluxes of precipitation and ET. The differences of less than 15% between model simulated ET and discharge, and the estimates from the validation datasets,

suggests that the water budget components are sufficiently well constrained to enable evaluation of the impact of climate warming on runoff and river discharge in Arctic rivers. In general, the comparisons with observations support the model's ability to reliably simulate key hydrological variables of interest.

# 4 Alterations connected to hydrological cycle intensification and permafrost thaw

## 4.1 Air temperature

In this analysis we use the simulations forced by the two climate models to bracket changes likely to occur this century, focusing primarily on twenty-year periods representing early (2000–2019) and late (2080–2099) century conditions. The IPSL model projects stronger warming compared to MPI, with warming between early and late century of 7.2 °C (domain-wide mean value) and 6.2 °C, respectively (Table 4). Both show the strongest warming over the highest latitudes of the pan-Arctic basin, with warming of over 10 °C across far northern Canada projected by IPSL. More modest warming of 3–4 °C is noted over southwestern Canada and central Eurasia in the MPI data.

Table 4: Climatological averages for early (2000–2019) and late (2080–2099) century periods from the simulations forced with IPSL and MPI meteorological data. [a]Relative (percentage) difference shown for each except air temperature, which is shown in degrees C. Differences are statistically significant for all quantities listed based on the paired T test (Sect. 2.4).

| Variable | PWBM-IPSL | | | PWBM-MPI | | |
|---|---|---|---|---|---|---|
| | early | late | % diff[a] | early | late | % diff[a] |
| air temp (C) | −5.3 | 1.9 | 7.2 | −5.3 | −0.9 | 6.2 |
| precipitation (mm yr$^{-1}$) | 578 | 697 | 21 | 573 | 643 | 12 |
| net precipitation (mm yr$^{-1}$) | 258 | 315 | 22 | 259 | 300 | 16 |
| rainfall (mm yr$^{-1}$) | 334 | 437 | 31 | 354 | 413 | 17 |
| snowfall (mm yr$^{-1}$) | 244 | 260 | 7 | 219 | 230 | 5 |
| rainfall fraction (%) | 56 | 62 | 11 | 43 | 63 | 47 |
| runoff (mm yr$^{-1}$) | 264 | 329 | 25 | 266 | 310 | 17 |
| F$_{sub}$ (%) | 27 | 35 | 30 | 30 | 34 | 13 |

In the results that follow, unless otherwise noted, statements reporting two statistics will be written in order for PWBM-MPI and PWBM-IPSL respectively. In nearly every instance, changes are greater with the latter simulation due to the influence of forcing from the more strongly warming (and wetter) IPSL climate model.

## 4.2 Precipitation

Rainfall rates have also been increasing across much of the pan-Arctic. Rainfall will continue to increase this century, particularly along favored storm track regions over northwestern Eurasia and western Alaska (Fig. 3a, S5a), where the majority of water vapor transport into the Arctic occurs (Nash et al., 2018). Climatological average rainfall (domain average) is higher by late century, with relative differences of 17 and 31% for the MPI and IPSL models, respectively (Table 4). Snowfall is projected to increase over a smaller geographic extent, mainly the higher latitudes and across the colder parts of eastern Eurasia, and decrease over most of the pan-Arctic, most prominently western Eurasia and southern Canada (Fig. 3b, S5b). The domain-wide change averages 5 and 7%. The sizable rainfall increases drive the projected rise in the fraction of rainfall to total precipitation (Fig. 3c, S5c) averaging 11 and 47% for the two simulations. Net precipitation—the difference between precipitation and the sum of evapotranspiration and snow sublimation—is projected to increase across most ($> 75\%$) of the pan-Arctic basin. Decreases will occur across southern Canada and Eurasia. For areas with and without permafrost, mean changes (increases) are 31 and 42%, and 5 and 6%, respectively. The simulations thus reveal bigger impacts—a net wetting—over permafrost regions, and a strong latitudinal south-north gradient in future precipitation changes that will influence river discharge quantity and quality.

## 4.3 Permafrost extent and active layer thickness

Research studies have documented hydrological cycle intensification and permafrost thaw across the terrestrial Arctic. To better understand changes in permafrost hydrology attributable to warming and increasing soil thaw we calculated ALT averages from the two climate-model-forced simulations (Fig. 4, S6). For PWBM-IPSL, permafrost area decreases by 7.8 million $km^2$ (12.3 to 4.5 million $km^2$) from the early to late century periods, a decline of 63% of present day permafrost area. For PWBM-MPI, some 4.9 million $km^2$ or 42% of present area loses permafrost (11.7 to 6.8 million $km^2$). Predictions of soil temperature from CMIP5 models point to permafrost fractional losses by end of century of 15% to 87% for RCP4.5, and 30%

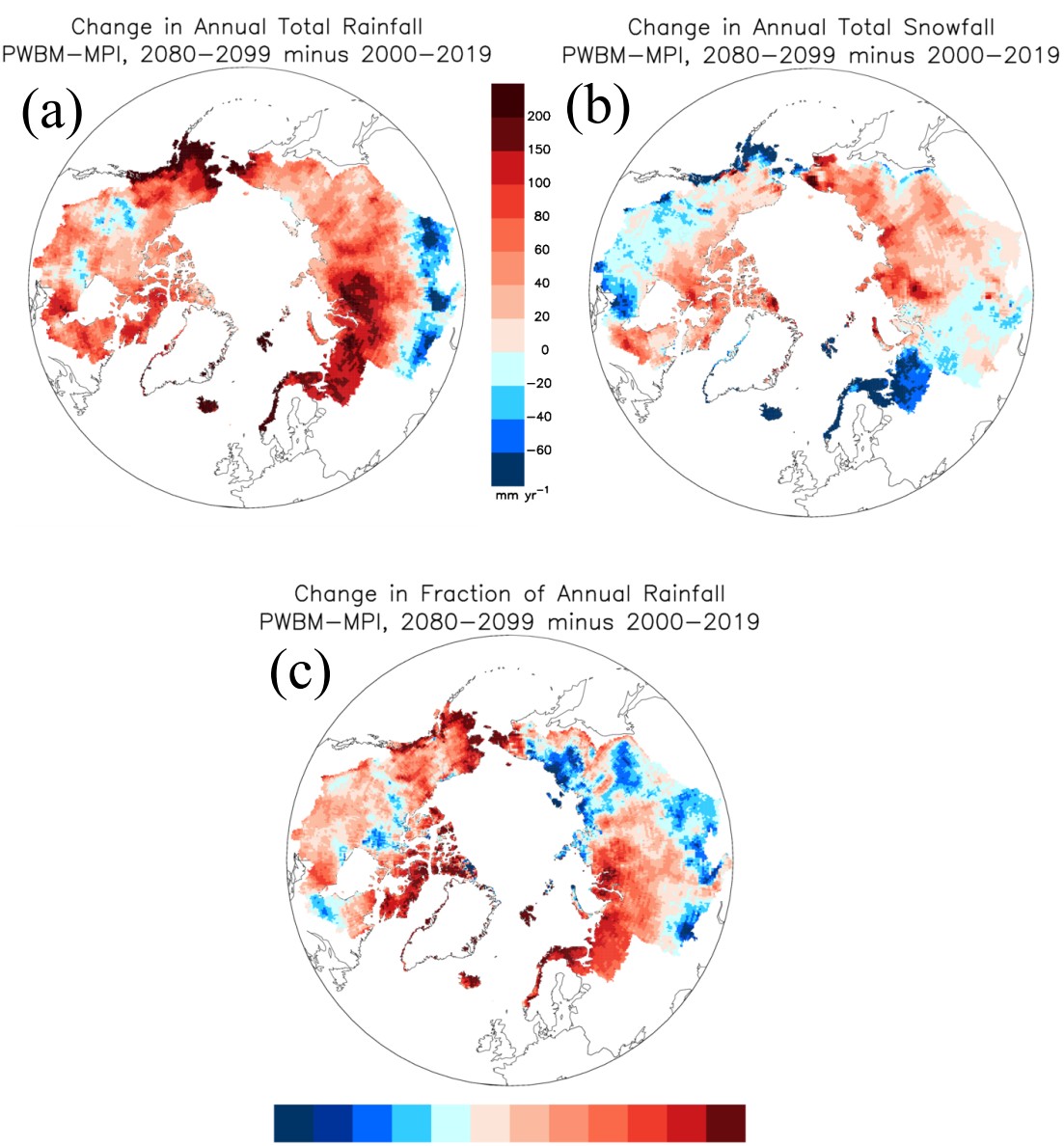

Figure 3: Change in (a) annual rainfall (mm yr$^{-1}$), (b) snowfall (mm yr$^{-1}$), and (c) the fraction of rainfall to total precipitation from PWBM-MPI simulation.

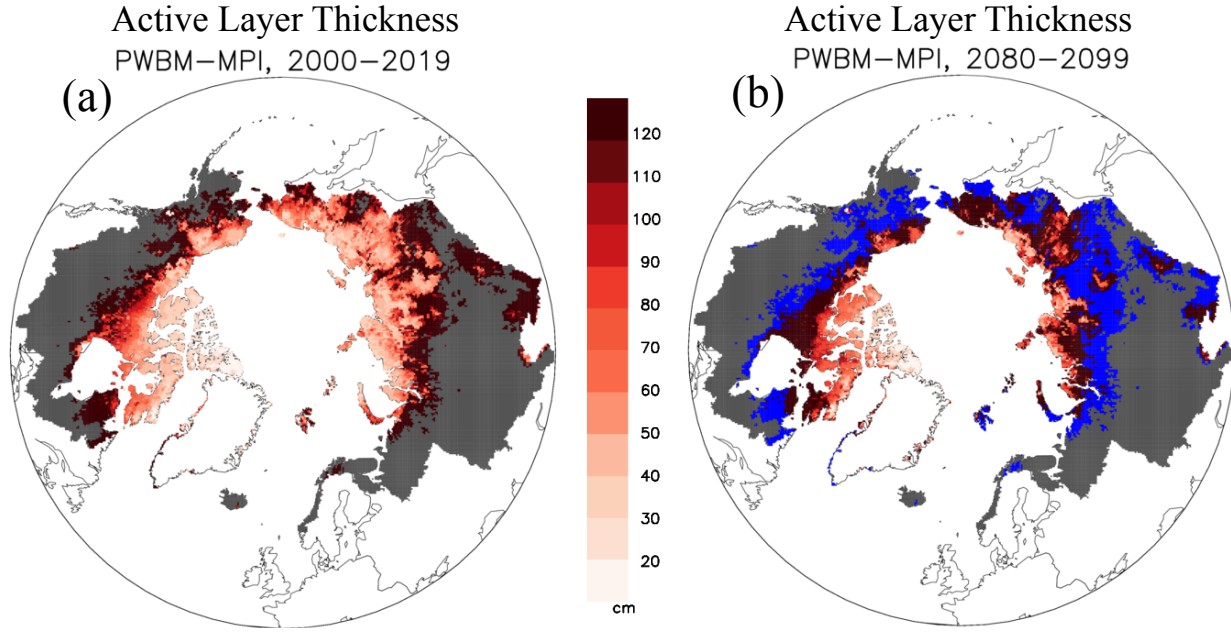

Figure 4: Simulated active-layer thickness (ALT, cm) for (a) early (2000–2019) and (b) late century (2080–2099) periods from PWBM-MPI. Blue shading highlights areas that are no longer characterized as permafrost in the future period. Gray areas are non-permafrost areas of the Arctic basin.

to 99% for RCP8.5 (Koven et al., 2013). Across areas that maintain permafrost, the ALT increases between the two periods average 56 and 91 cm. For comparison, estimates over permafrost areas obtained from an air temperature-based thawing index applied to 16 CMIP5 models (2006–2100) forced under RCP8.5 averaged a similar 6.5 cm decade$^{-1}$.

## 4.4   Runoff and river discharge

Annual runoff within the pan-Arctic basin is typically highest across eastern Canada, western Eurasia, and coastal regions of western Canada and western Alaska. Runoff changes between the early and late century periods were calculated here to assess future alterations to river discharge (Fig. 5a, S7a). In Eurasia the change in annual total runoff, as a percent of the early period, is greater over northeast parts of the continent. Across North America the increases are also greater in the colder northern parts of the Canadian archipelago and over northern Alaska. Averaged across all grid cells, annual runoff increases by 19% (45 mm yr$^{-1}$) and 31% (65 mm yr$^{-1}$) from PWBM-MPI and PWBM-IPSL, respectively. Not surprisingly, the spatial pattern in runoff change closely aligns with the pattern in net precipitation. There is also a significant difference in the mean change in annual runoff between grid cells with permafrost (67 and 99 mm yr$^{-1}$ increase) and those without permafrost (21 and 25 mm yr$^{-1}$). This divergence is driven by changes in net precipitation (64 and 89

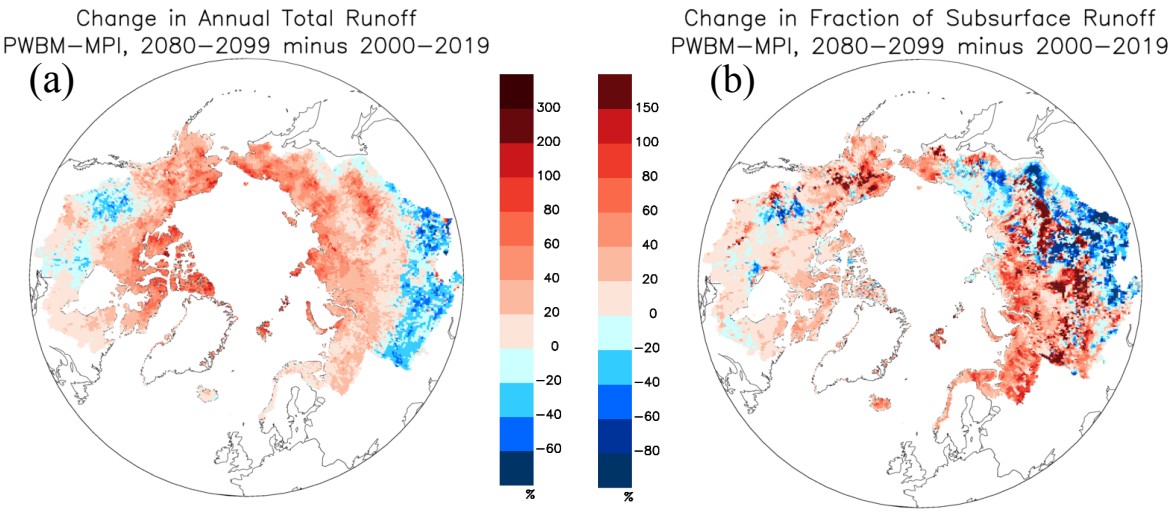

Figure 5: Change in (a) annual total runoff (%) and (b) fraction of subsurface to total runoff ($F_{sub}$, %) from the simulations.

mm yr$^{-1}$ vs. 18 and 19 mm yr$^{-1}$), as well as differing influences from deepening ALT and longer thawed periods in areas with and without permafrost. Across permafrost areas, the difference between net precipitation and runoff—in a water budget, an approximation for change in storage—is 3–10 mm yr$^{-1}$, a small amount relative to the runoff increase. Over the early century period, river discharge volume is 5839, 5955, 5917 km$^3$ yr$^{-1}$ for the PWBM-W5E5, PWBM-MPI, PWBM-IPSL simulations respectively (Fig. S4). By late century, discharge volume increases to 6955 and 7374 km$^3$ yr$^{-1}$, relative increases of 17 and 25% for PWBM-MPI, and PWBM-IPSL respectively (runoff equivalents in Table 4). The trend is statistically significant (p < 0.01) for both time series.

A transition from runoff dominated by surface water contributions toward increasing amounts of subsurface flow is expected as the climate warms (Frey and McClelland, 2009). Compared to change in total runoff, the change in the fraction of subsurface to total runoff ($F_{sub}$) is more spatially variable across the pan-Arctic (Fig. 5b, S7b). During the early century period, $F_{sub}$ averages 30% and 27% in the PWBM-MPI and PWBM-IPSL simulations respectively (Fig. 6). The fractions increase to 34% and 35% by end of century, giving relative (percent) increase in domain mean $F_{sub}$ of 13 and 30% for PWBM-MPI and PWBM-IPSL respectively. Based on the modest warming PWBM-MPI run, approximately 72% of permafrost areas will

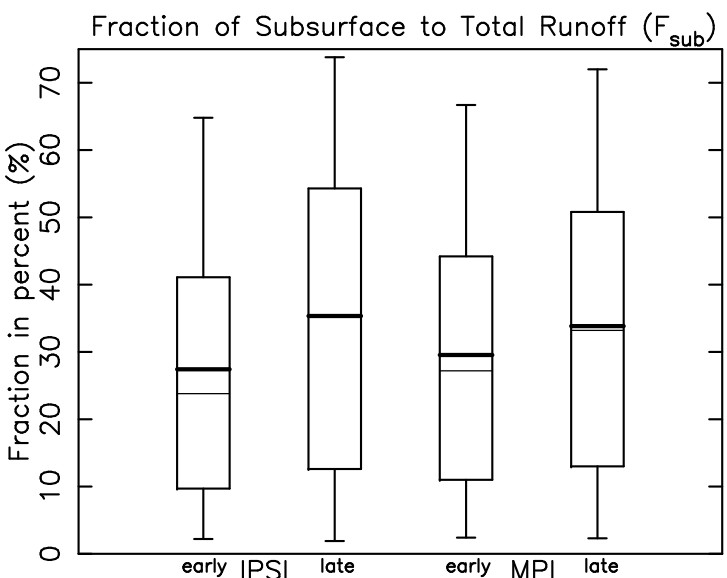

Figure 6: Fraction of subsurface to total runoff ($F_{sub}$) for early and late century periods for all pan-Arctic grids from PWBM-IPSL and PWBM-MPI simulations.

have higher subsurface runoff fractions by end of century. This spatial extent increases to 88% of the permafrost region under the more aggressive warming depicted under PWBM-IPSL (Fig. S7b). The shift in $F_{sub}$ is larger in permafrost areas, with significant differences in spatial mean $F_{sub}$ in areas with and without permafrost (relative differences 15.7 and 13.5% respectively for PWBM-MPI; 31.1 and 24.4% for PWBM-IPSL). The PWBM-MPI simulation reveals a significant relationship (p < 0.01) between change in ALT and F$_{sub}$, with a 6.4% increase in $F_{sub}$ per 0.1 m increase in ALT. While the positive correlation does not exist under PWBM-IPSL, the more pervasive growth in $F_{sub}$ in PWBM-MPI suggests a connection between soil thaw and increasing contributions from subsurface runoff to river discharge during this century, particularly in regions underlain by permafrost.

The runoff changes in both simulations exhibit a significant positive relationship with latitude (Fig. 7a, S8a). The linear fit suggests an additional 2.9 and 4.2% runoff (PWBM-MPI and PWBM-IPSL) for each degree northward in latitude. Under this pattern river discharge shifts over time to being sourced more from the northerly parts of the four largest river basins (Ob, Yenesey, Lena, Mackenzie; Fig. 8a, S9a, Table 5). Decreases are projected for the southerly half of the Ob, Yenesey, and Mackenzie Rivers. For the Ob basin, less runoff across the southern half of the river basin will be offset by higher flow in the north, so that annual total discharge exported

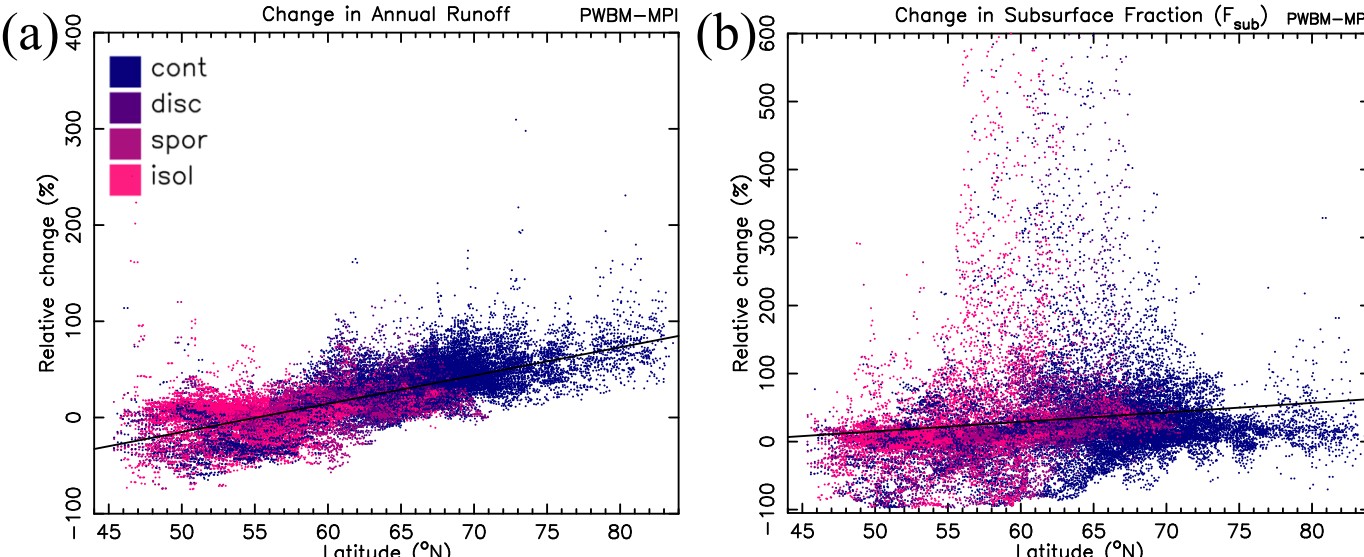

Figure 7: Change in (a) annual total runoff (%) and (b) $F_{sub}$ with grid cell latitude from PWBM-MPI simulation for all pan-Arctic domain grid cells. Colors indicate permafrost classification (continuous, discontinuous, sporadic, or isolated) for the cell from IPA dataset (Figure 1c).

at the coast is relatively unchanged. The Yenesey shows a similar pattern, with accumulated discharge at the coast higher by late century. The Lena and Mackenzie Rivers will receive substantial additional discharge from their northern areas, with the Lena projected to export 66 and 128 $km^3$ $yr^{-1}$ (16 and 31%) more freshwater discharge by late century. The sharp increase in export from the Yenesey and Lena arising from their northern watersheds is driven primarily by higher snowfall rates (Fig. 3b, S5b). Averaged across the four, the downstream half of the rivers will receive approximately 20–30% more accumulated discharge from the northern half of their contributing area. A south-north gradient also exists in soil carbon storage in these basins, with the highest amounts in the far north (Fig. 8b, S9b). Subsurface runoff increases are also greater to the north (Fig. 7b, S8b), though the scatter is substantial compared to the change in annual total runoff.

Runoff is projected to increase during most months in both simulations (Fig. 9, S10), with monthly changes remarkably similar between the two runs. Averaged over seasons, runoff increases (depth in mm) are greatest in spring (MAM). The increase in spring, particularly during May, is attributable to additional snowmelt runoff and a shift to earlier snowpack melting. As a consequence, less snowmelt and runoff occur in June. Averaged across the six largest rivers (Ob, Yenesey, Lena, Mackenzie, Yukon, Kolyma), peak daily discharge at each coastal outlet shifts earlier by end of

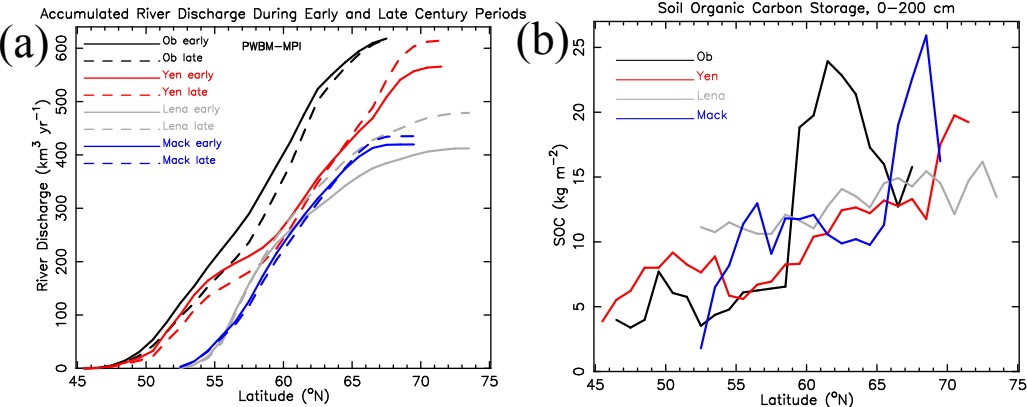

Figure 8: (a) Accumulated annual total river discharge (km³ yr⁻¹) for the Ob, Yenesey, Lena, and Mackenzie Rivers for 1° latitude bands as averages over early (solid line) and late (dashed) century periods from PWBM-MPI. (b) Soil carbon storage (kg m⁻²) in soil 0–200 cm zone from the Northern Circumpolar Soil Carbon Database (Hugelius et al., 2013).

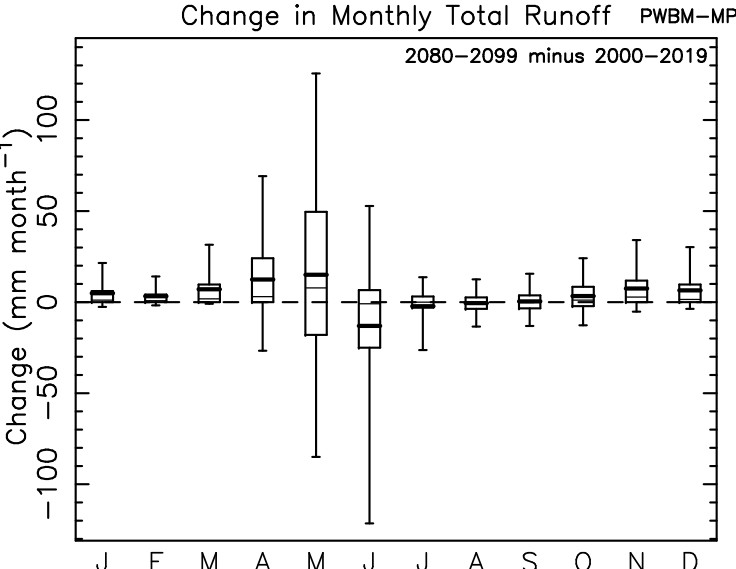

Figure 9: Distribution in change in monthly total runoff (mm month⁻¹) between early and late century periods for all pan-Arctic grid cells from PWBM-MPI.

Table 5: Relative (percentage) change in accumulated river discharge for the upstream (southern) half and downstream (northern) half of each of the four largest Arctic rivers. Averages are calculated from the totals shown in Fig.s 8, S7. Total row represents the average from the four.

| | PWBM-IPSL | | PWBM-MPI | |
| --- | --- | --- | --- | --- |
| River | up (%) | down (%) | up (%) | down (%) |
| Ob | −9.8 | 7.4 | −19.4 | 13.6 |
| Yenesey | −1.5 | 27.9 | −14.2 | 22.2 |
| Lena | 26.4 | 43.8 | 12.5 | 25.9 |
| Mackenzie | −0.2 | 35.3 | −5.3 | 17.3 |
| Total | 3.7 | 28.6 | −6.6 | 19.7 |

century by approximately 11 days in both simulations (DOY 180 to 169 in PWBM-IPSL and DOY 176 to 165 in PWBM-MPI). Runoff is largely unchanged in July, August and September, and the changes are not statistically significant in June and July due to the high degree of spatial variability. Seasonally, the relative change (percentage change) is greatest in winter, with runoff by late century a factor of 5–10 greater compared to the early century period averages. Significant percentage increases are noted in autumn and spring as well. Interestingly, snow storage (snow water equivalent, SWE) increases in both simulations are significant in February, March, and April only. Notably, no increase in SWE is projected during autumn.

The intensifying hydrological cycle and thawing permafrost will manifest in changing amounts of surface and subsurface runoff contributions to river discharge (Fig. 10). The shifts vary strongly with season, and spatially across the terrestrial Arctic, with remarkably similar change magnitudes in the two simulations, due largely to similarities in patterns in net precipitation and its change this century. At the pan-Arctic scale, modest increases are projected in both surface and subsurface runoff for the annual total and in winter, spring, and autumn. The acceleration during winter and autumn will come predominantly from additional subsurface runoff. Spring increases are mainly attributable to increased surface runoff. Runoff is projected to decrease slightly in summer due to less surface runoff, despite a small increase in subsurface runoff. The autumn change is particularly noteworthy over northern Alaska. Also there, summer shows a strong shift from surface to subsurface runoff. Runoff decreases are projected to occur in most seasons over southwest Canada, owing to relatively large precipitation declines (Fig. 3, S5).

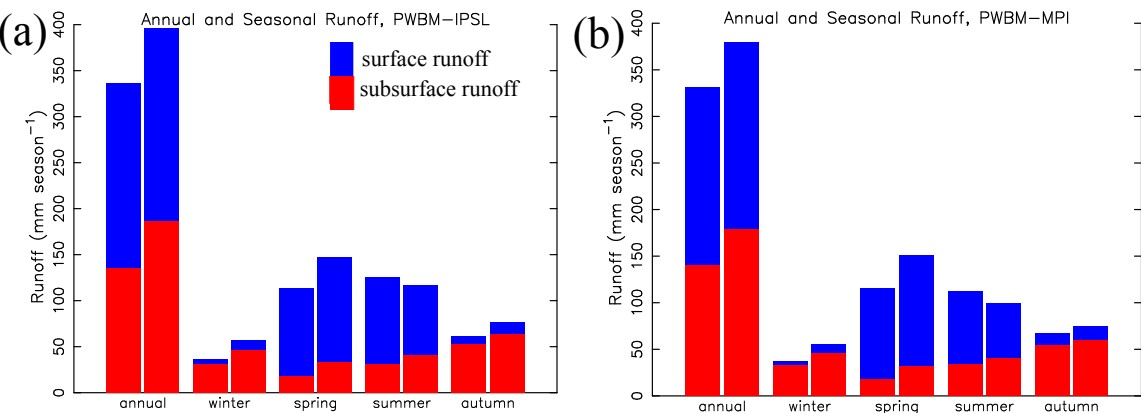

Figure 10: Annual and seasonal total runoff for the early (left bar) and late century (right bar) periods, expressed as surface (blue) and subsurface (red) amounts for (a) PWBM-IPSL and (b) PWBM-MPI simulations.

# 5    Discussion

The Arctic basin is drained by several rivers that receive runoff contributions over great distances, from grasslands and forests in the south to tundra in the north. Surface runoff has typically been a substantial component of river discharge, with subsurface flows characterizing low flows in summer and early fall. These characteristic patterns and dynamics are shifting due to influences from warming, primarily hydrological cycle intensification and permafrost thaw. The shifts are altering the water cycle from processes manifesting both horizontally, via primarily atmospheric effects, and vertically, from soil thaw, and seasonally, through a combination of both impacts. Recent research suggests that a warming Arctic will experience changes in moisture sources that will influence freshwater exports from rivers. The two coupled climate models from which outputs were used in this study capture substantial precipitation increases in regions adjacent to the Arctic Ocean. This is a robust feature of climate models that is linked to a more open Arctic Ocean later this century (Barnhart et al., 2016; McCrystall et al., 2021). River basins near the western Arctic Ocean, particularly far northeast Eurasia, northwest Canada, and northern Alaska, will experience relatively large increases in river discharge, driven partly by higher snowfall rates and spring SWE amounts. These are cold areas that will warm significantly and, in turn, increasingly be fed by additional moisture, including from more frequent atmospheric rivers (Zhang et al., 2023). In contrast, southern parts

of the pan-Arctic basin are projected to experience a decline in net precipitation and runoff contributions to rivers. In general, rivers in central Eurasia and southern Canada will receive less runoff, particularly during summer. Our results suggest that nearly 90% of the increase in river discharge from permafrost regions will arise from an increase in net precipitation (Cubasch et al., 2001), rather than a "de-watering" of permafrost from thawing soil ice, which likely also played a smaller role over the $20^{th}$ century (McClelland et al., 2004). This connection to net precipitation is consistent with attribution studies for the river discharge trends observed during the recent past (McClelland et al., 2004, 2006; Zhang et al., 2013). Our results point to significant shifts in sources of freshwater entering Arctic rivers, with less runoff to river networks in the south and more in the north. The headwaters of the large Arctic rivers like the Lena, Ob, Yenisey, Mackenzie, originate well south of what is typically considered Arctic lands. The simulations suggest that by end of century, some 20–30% more freshwater discharge will enter, accumulate in, and be export from the northern half of the four large rivers.

In addition to geographic shifts involving atmospheric influences, ongoing soil thaw and permafrost losses will also influence runoff and materials contributions to rivers. Our results support a growing body of evidence that deepening active layers and losses in permafrost extent will increase subsurface runoff contributions to rivers. Permafrost extent declines by 42 and 63% (PWBM-MPI and PWBM-IPSL respectively) between early (2000–2019) and late (2080–2099) century periods, indicative of recent and future permafrost degradation. Recent observations in northern Alaska suggest that increased precipitation and deepening ALT play increasingly important roles in sustaining low flows and enhancing subsurface hydrologic processes (Arp et al., 2020; Cooper et al., 2023). Projected changes in subsurface runoff are more spatially variable compared to total runoff, though a similar south-north gradient exists. Increased subsurface runoff can lead to decreases in summer stream temperatures in headwater catchments (Sjöberg et al., 2021). Pronounced seasonal shifts in runoff contributions will also occur. Increased runoff in late spring will likely be driven by higher snow storage and earlier melt that will shift peak spring freshet runoff earlier by approximately 11 days this century. Increased autumn discharge in the simulations is not attributable to higher SWE, forced instead by thawing permafrost that is lengthening the period when flow occurs, and creating deeper active layers that store and release water later in the season. More runoff during November and December, an approximate 5-fold increase in the modest warming simulation, highlights the physical connection between warming, permafrost degradation, and increasing subsurface flows to streams and rivers (St. Jacques and Sauchyn, 2009; Rawlins et al., 2019). The relatively large changes in November–April runoff de-

scribed here are congruent with a recent study that documented a 10% per decade increase in cold season discharge from nine rivers in Alaska with long data records (Blaskey et al., 2023). Warming, prominent in this region during autumn and early winter, can promote increased soil water storage, delaying the release of water into the streams, and thus contribute to increases in winter flow (Streletskiy et al., 2015). Results of this study support the hypothesis that across the Arctic basin subsurface runoff increases will be greatest in permafrost areas.

Taken together along with other studies eg. (Mann et al., 2022; Tank et al., 2023), the spatial shifts suggest alterations in materials exported to coastal waters. Warming and higher rainfall rates will enhance thaw and increase coastal erosion. Higher runoff rates will drive additional subsurface contributions of freshwater and DOC to coastal seas and lagoons (Connolly et al., 2020). More cold season river discharge has the potential to affect sea ice dynamics and other near-shore processes involving quantities such as salinity and biogeochemistry. The impacts extend to water quality and materials exports by rivers. For example, DOC input to the Arctic Ocean has a very high temporal and geographical variability with a strong bias towards the large Eurasian Rivers and the freshet period (Amon et al., 2012). Our results suggest impacts to carbon of differing quality, as Amon et al. (2012) reported that lignin phenol and p-hydroxybenzene composition of Arctic river DOC point to the abundance of young, boreal-vegetation-derived leachates during spring flood and older, soil-, peat-, and wetland-derived DOC during groundwater dominated low ow conditions. In northern tundra areas where soil carbon amounts are greater, warmer temperatures and increased runoff will likely lead to increased riverine DOC exports. Indeed, Frey and Smith (2005) concluded that, assuming no change in either river discharge or in-channel processes, warming would produce a 2.7–4.4 Tg $yr^{-1}$ increase in terrestrial DOC flux from West Siberia to the Arctic Ocean by 2100, with even larger increases likely should river discharge from the region continue to increase, as depicted in the simulations examined here. Warming and shifting snowmelt dynamics could increase transport and mobilization of DOC as subsurface pathways become active earlier in the year (Croghan et al., 2023). In contrast, some areas may experience a decrease in DOC export over time due to longer flow paths and residence times, along with increased microbial mineralization of DOC in the soil column (Striegl et al., 2005). Increasing soil thaw is expected to accelerate the release of old carbon (Dean et al., 2018; Schwab et al., 2020), which in turn will be entrained into, processed by, and exported from Arctic rivers. Moreover, DOC from deep sediments ($> 3$ m) could also become a significant contribution of carbon to Arctic rivers as the climate continues to warm (Mohammed et al., 2022). Nitrate concentrations are greater at lower latitudes as compared with higher latitudes where

permafrost is more prominent (Frey and McClelland, 2009). Changes expressed predominantly across northern parts of the Arctic basin will have a direct influence on coastal zone processes. On balance, our results point to continued increases in DOC export by Arctic rivers, and the mobilization and transport of ancient carbon in subsurface runoff from permafrost areas.

The use of two climate model forcing sets increases confidence in elements of the model outputs and associated analysis. It is noteworthy that results involving runoff, in particular the spatial patterns, are similar between the two simulations. Magnitudes of air temperature and precipitation increases are greater in the simulation forced with IPSL (PWBM-IPSL). Under those warmer temperatures, the Hamon potential evapotranspiration function captures the temperature dependence on actual and potential evapotranspiration. Higher precipitation rates in a warmer forcing scenario, like IPSL, are offset by higher simulated ET, resulting in relatively similar magnitudes of annual net precipitation and annual total runoff. This plausible modeling result suggests less uncertainty with the magnitudes of runoff changes compared with the changes in meteorological forcings projected by the climate models. The model validation analysis suggests that the magnitude of simulated annual total runoff and discharge are comparable to independent observational datasets, with time trends similar in magnitude to those reported in other studies.

Salient conclusions from this study come with caveats related to the limits of the analysis. Foremost is the large degree of uncertainty in meteorological data across Arctic regions, attributable to a sparse observation network, as well as uncertainties in the magnitude of meteorological changes projected by the two coupled climate models. This uncertainty is ameliorated somewhat through the use of reanalysis data and model calibration. Results are implicitly linked to the connection between landscape runoff and river discharge export. Results are also influenced by the choice of climate model forced under the SSP3-7.0 scenario. In light of this, one might expect lower magnitudes of change should atmospheric greenhouse gas concentrations not rise to levels depicted in SSP3-7.0. The broad spatial extent and moderate model resolution ($25 \times 25$ km grid cells) employed in this study limit our ability to incorporate influences such as thermokarst and talik formation on runoff contributions to streams and rivers. However, it is not clear that these local processes are a major component of riverine materials exports by Arctic rivers (Dean et al., 2018). The model simulations do not include interactions between lakes and the river networks, so, impacts from lake thaw drainage events (Smith et al., 2005; Andresen and Lougheed, 2015; Jones et al., 2022) are not simulated. The influence of land subsidence on soil temperature, moisture, and water storage is also not simulated. While subsidence is unlikely to lead to abrupt thaw over large areas, it can

have significant effects on the hydrology of polygonal tundra, generally increasing landscape runoff (Painter et al., 2023). The effect on large river basins will depend on the fraction of those basins that contain polygonal tundra. Our results underscore the importance in better understanding the myriad transformations reshaping Arctic environments. Large changes in the far north emphasize the need for more frequent and spatially extensive sampling of small and medium-sized rivers that ring the Arctic Ocean. Increased confidence in the magnitude of likely responses will require a multi-model, multi-scenario ensemble of simulations to obtain a range of projections consistent with known uncertainties. Incorporating small-scale effects such as thermokarst and lake drainage on river discharge will require higher-resolution simulations. New model parameterization obtained from high resolution remote sensing observations will improve model capabilities in simulating permafrost hydrology in data sparse regions of the Arctic.

# 6 Code and data availability

The W5E5 data are available at https://dataservices.gfz-potsdam.de/pik/showshort.php?id=escidoc:4855898 (last access: 15 October 2022). The MERRA reanalysis data are available at https://gmao.gsfc.nasa.gov/reanalysis/MERRA/ (last access: 23 January 2023). The ECMWF Reanalysis v5 (ERA5) data are available at https://www.ecmwf.int/en/forecasts/dataset/ecmwf-reanalysis-v5 (last access: 19 March 2023). The TPDC data are available at http://data.tpdc.ac.cn/en (last access: 3 February 2023). The IPA permafrost data in the Circum-Arctic Map of Permafrost and Ground-Ice Conditions, Version 2 are available at https://nsidc.org/data/ggd318/versions/2 (last access 1 August 2022). The Global Land Evaporation Amsterdam Model (GLEAM) data are available at https://www.gleam.eu/ (last access: 17 April 2023). The pan-Arctic ET data derived from remote sensing are available at http://files.ntsg.umt.edu/data/PA_Monthly_ET/ (last access: 16 April 2023). Climate model data used as forcings are available in the ISIMIP Repository located at https://data.isimip.org/. Model source code, forcings, parameterizations, and outputs are available at https://data.ess-dive.lbl.gov/datasets/ess-dive-9d7ec367652ba43-20240201T162009168

# 7 Author contributions

MAR set up and executed the simulations, analyzed the results, and wrote the initial draft. AVK prepared the modeling forcing datasets and contributed to the analysis writing of the accepted manuscript.

# 8 Competing interests

The authors declare that they have no conflict of interest.

# 9 Acknowledgements

The PWBM simulations were performed on high performance computing resources provided by the Massachusetts Green High Performance Computing Center. We thank John Kimball, James McClelland, Vladimir Alexeev, the two reviewers, and editor for comments which greatly improved the paper.

# 10 Financial support

This work was supported by funding from the U.S. Department of Energy, Office of Science, Office of Biological and Environmental Research (Grant No. DE-SC0019462), the National Aeronautics and Space Administration (Grant No. 80NSSC19K0649), and the National Science Foundation, Division of Polar Programs (Grant No. NSF-OPP-1656026).

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

*Supplemental Information for*

# Regime Shifts in Arctic Terrestrial Hydrology Manifested From Impacts of Climate Warming

1083 Michael A. Rawlins[1] and Ambarish V. Karmalkar[2,1]

1084 [1]Department of Earth, Geographic, and Climate Sciences, University of Mas-
1085 sachusetts, Amherst, MA 01003, USA
1086 [2]Department of Geosciences, University of Rhode Island, Kingston, RI 02881, USA
1087 *Correspondence to:* Michael A. Rawlins (mrawlins@umass.edu)

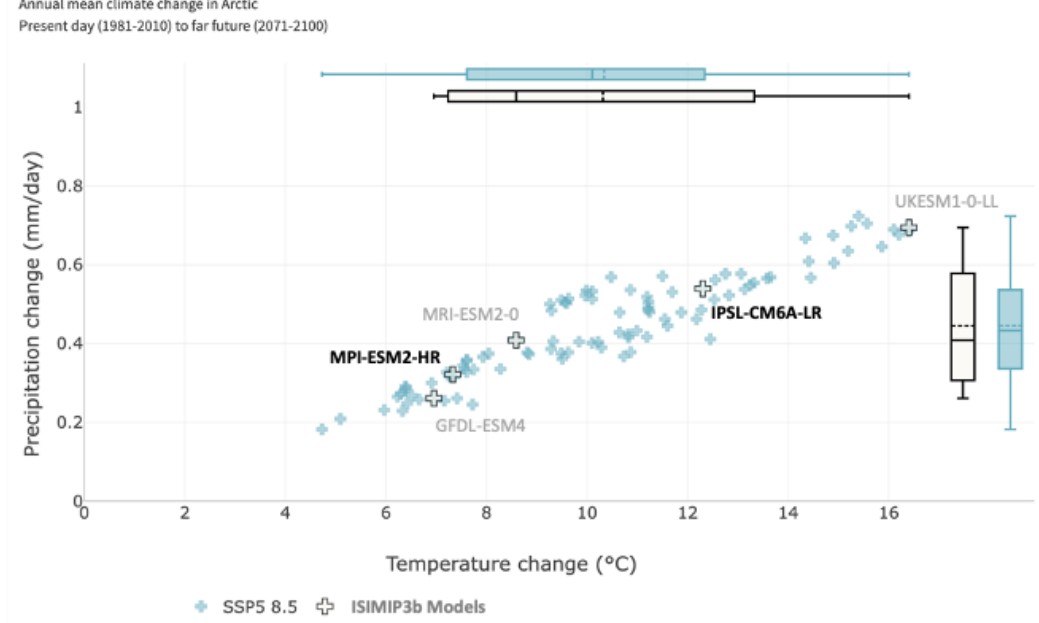

Figure S1: Projected changes in temperature (in °C) and precipitation (in mm day$^{-1}$) for 2070–2100 relative to 1981–2010 mean for the Arctic based on climate models in the CMIP6 archive. The projections are shown for SSP5-8.5. Five CMIP6 models included in ISIMIP3b are highlighted, with the two that were selected as climate inputs in this study shown in bold. Box and whiskers show ranges in temperature and projections spanned by the full CMIP6 ensemble (blue) and the five ISIMIP3b models (black). The figure was created using the GCMeval tool at https://gcmeval.met.no/

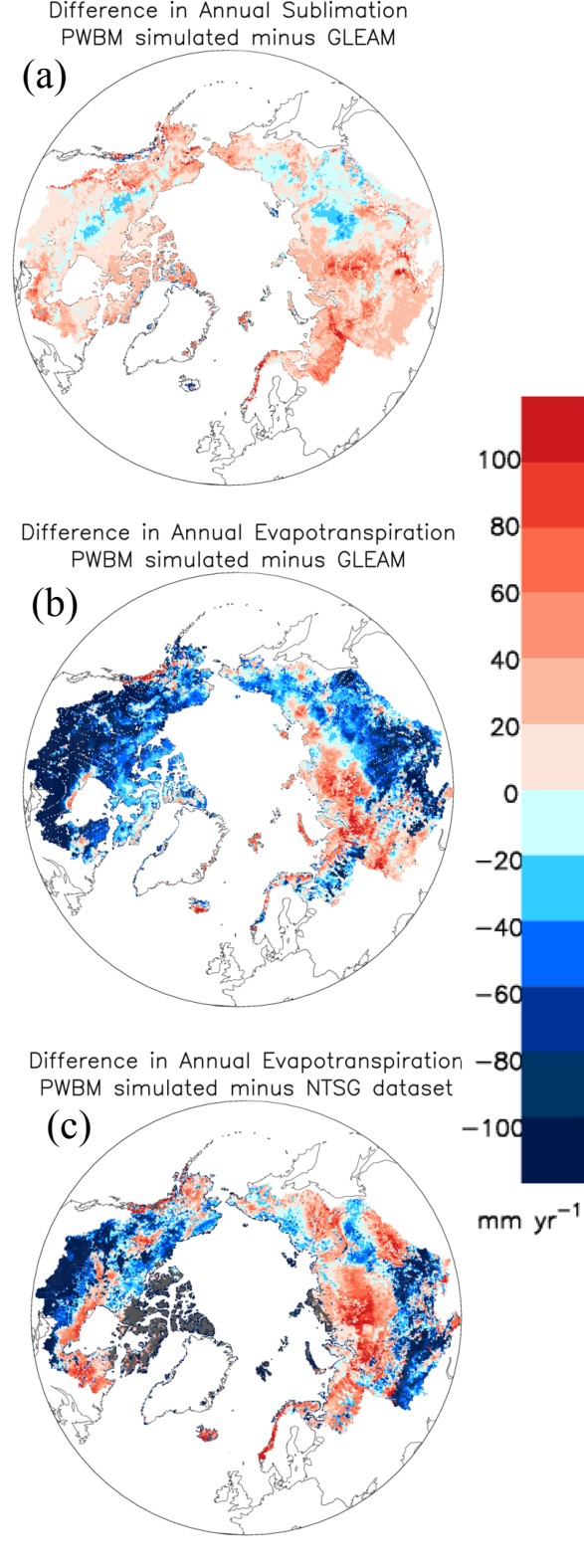

Figure S2: Difference in annual total sublimation (mm yr$^{-1}$) between simulations with PWBM forced with WFE5 and GLEAM dataset (a) and annual total ET (mm yr$^{-1}$) between PWBM and GLEAM (b), and difference between PWBM and a dataset made available by the Numerical Terradynamic Simulation Group at the University of Montana (c).

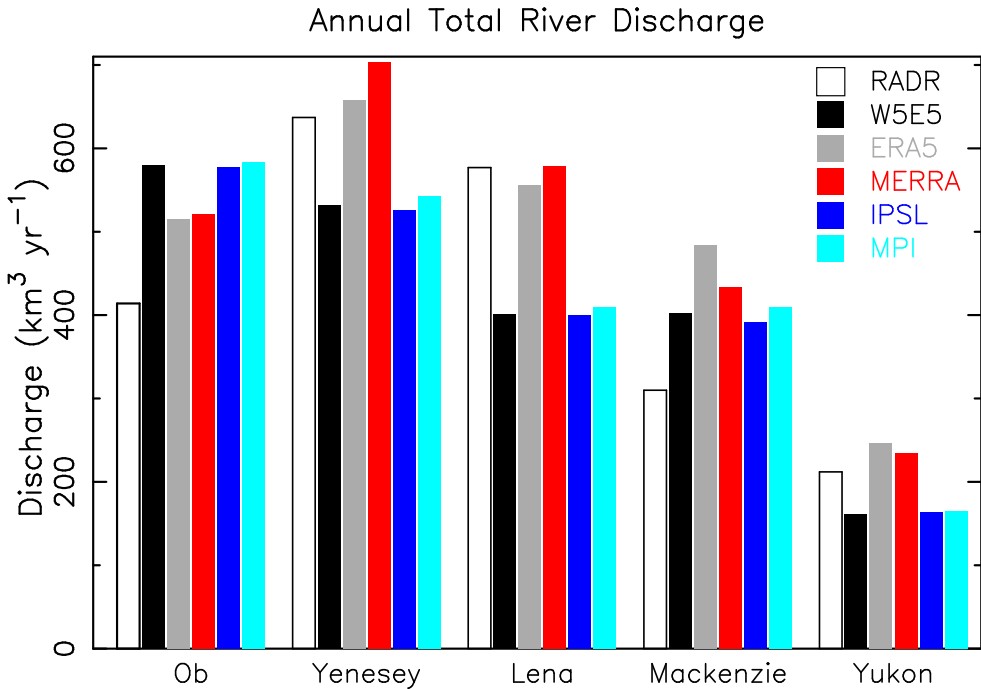

Figure S3: Annual total river discharge (km$^3$ yr$^{-1}$) for the five largest Arctic rivers. The RADR dataset (Feng et al., 2021) serves as validation for the simulated estimates (PWBM-). Discharge volume shown as an average over the period 1984–2018 for the RADR data, 1980–2019 for the simulations forced by W5E5, ERA5, IPSL, and MPI, and 1980–2013 for the simulation forced by MERRA.

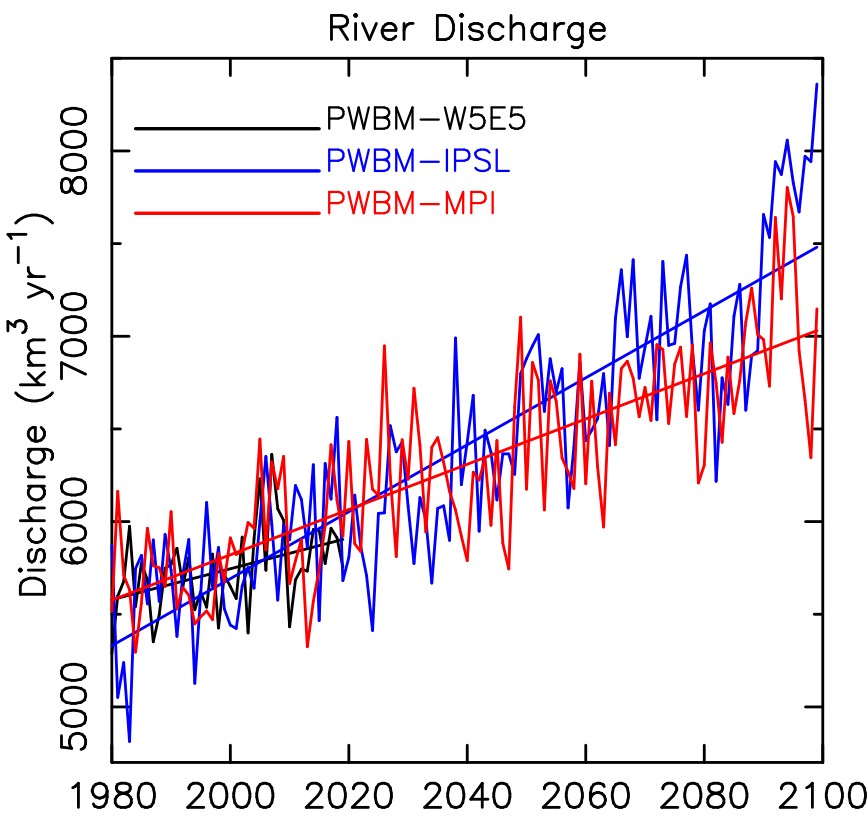

Figure S4: Annual total river discharge (km$^3$ yr$^{-1}$) from simulations for 1980–2019 and 1980–2100. Linear trend shown.

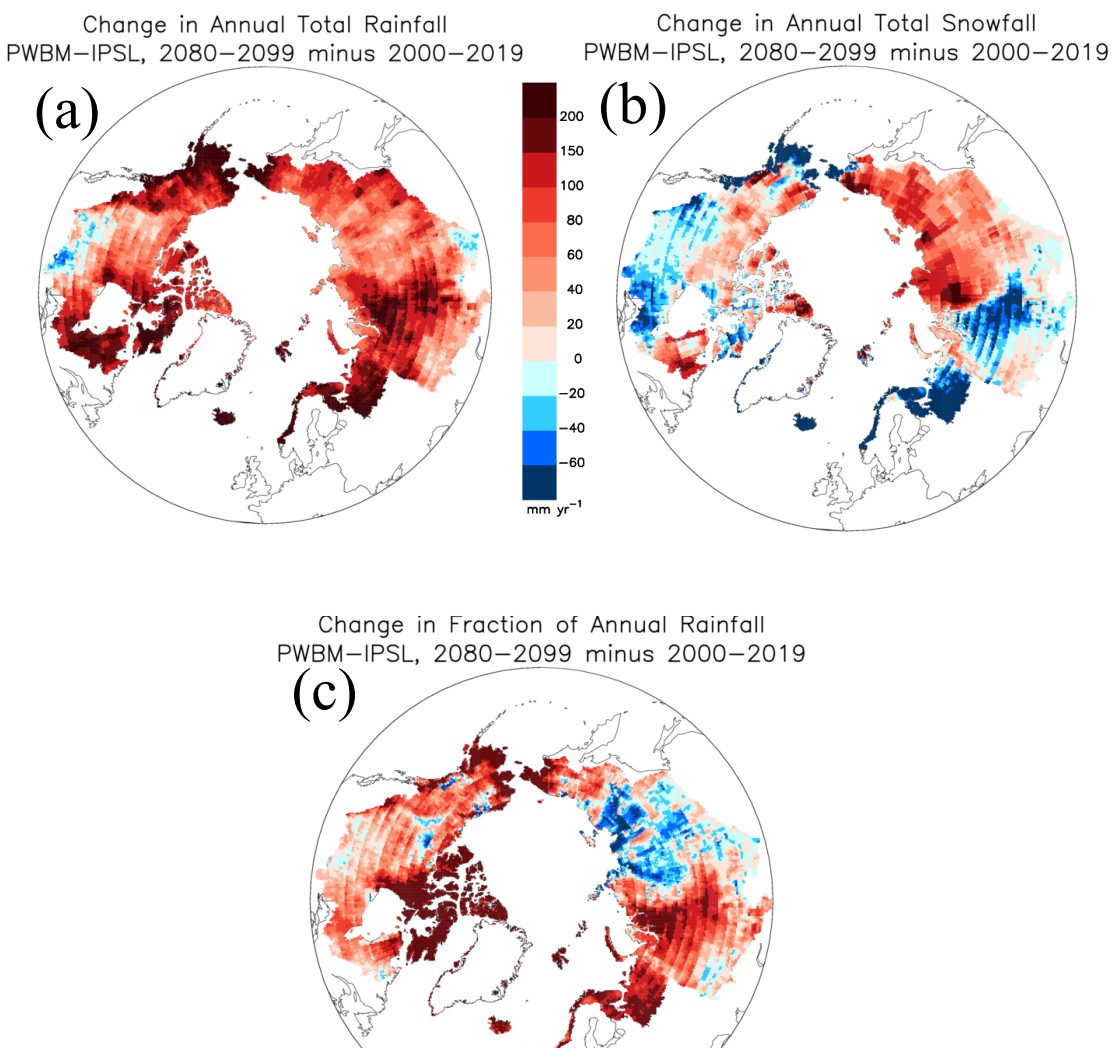

Figure S5: Change in (a) annual rainfall (mm yr$^{-1}$), (b) snowfall (mm yr$^{-1}$), and (c) the fraction of rainfall to total precipitation from PWBM-IPSL simulation.

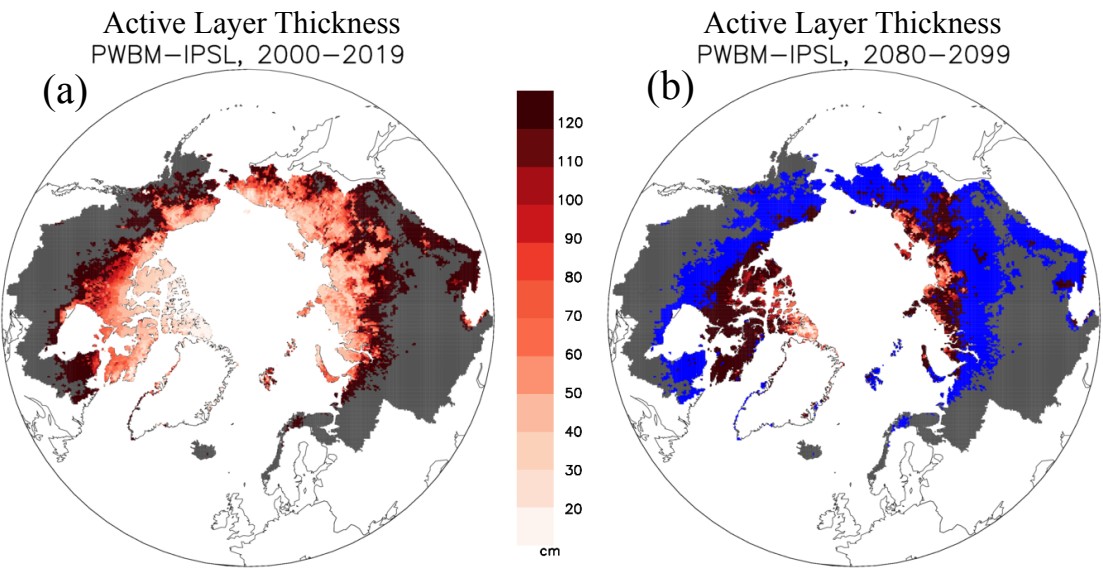

Figure S6: Simulated active-layer thickness (ALT, cm) for (a) early (2000–2019) and (b) late century (2080–2099) periods from PWBM-IPSL. Blue shading highlights areas that are no longer characterized as permafrost in the future period. Gray areas are non-permafrost areas of the Arctic basin.

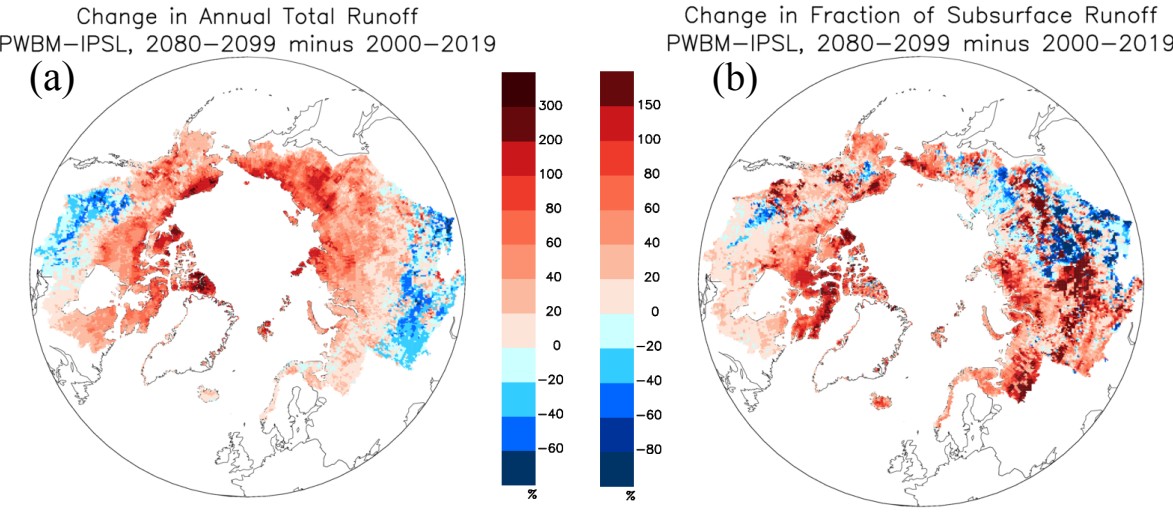

Figure S7: Change in (a) annual total runoff (%) and (b) $F_{sub}$ (%) from PWBM-IPSL.

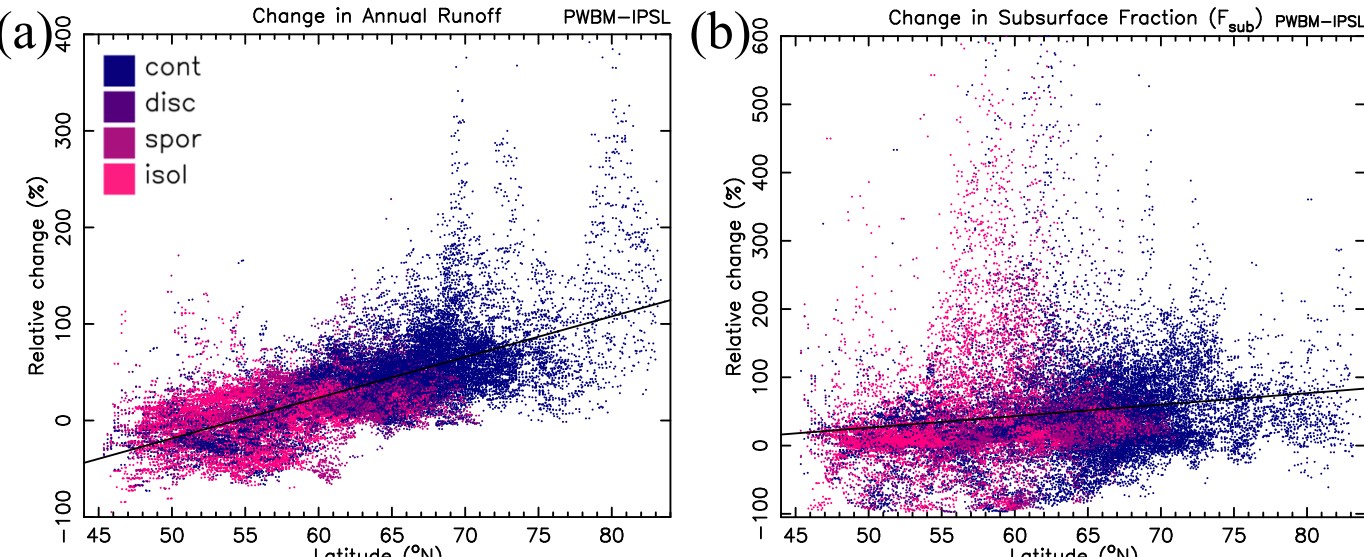

Figure S8: Change in (a) annual total runoff (%) and (b) $F_{sub}$ with grid cell latitude from PWBM-IPSL simulation for all pan-Arctic domain grid cells. Colors indicate permafrost classification (continuous, discontinuous, sporadic, or isolated) for the cell from IPA dataset (Fig. 1a).

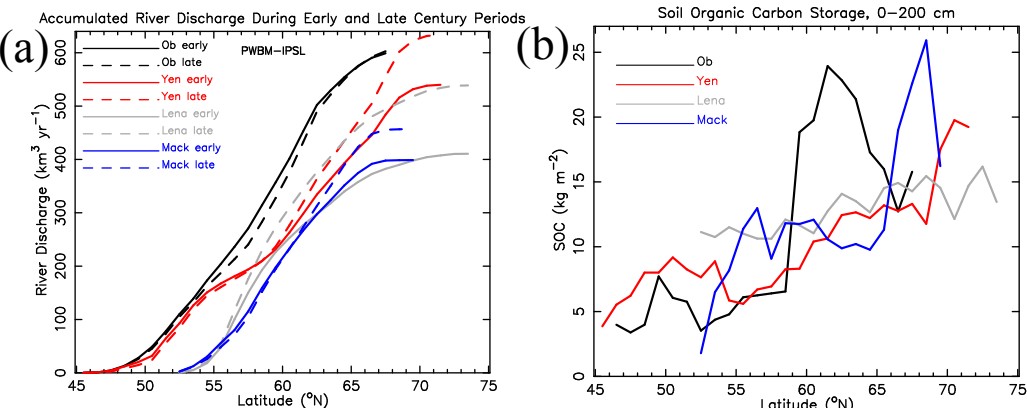

Figure S9: (a) Accumulated annual total river discharge (km$^3$ yr$^{-1}$) for the Ob, Yenesey, Lena, and Mackenzie Rivers for 1° latitude bands as averages over early (solid line) and late (dashed) century periods from PWBM-IPSL. (b) Soil carbon storage (kg m$^{-2}$) in soil 0–200 cm zone from the Northern Circumpolar Soil Carbon Database (Hugelius et al., 2013).

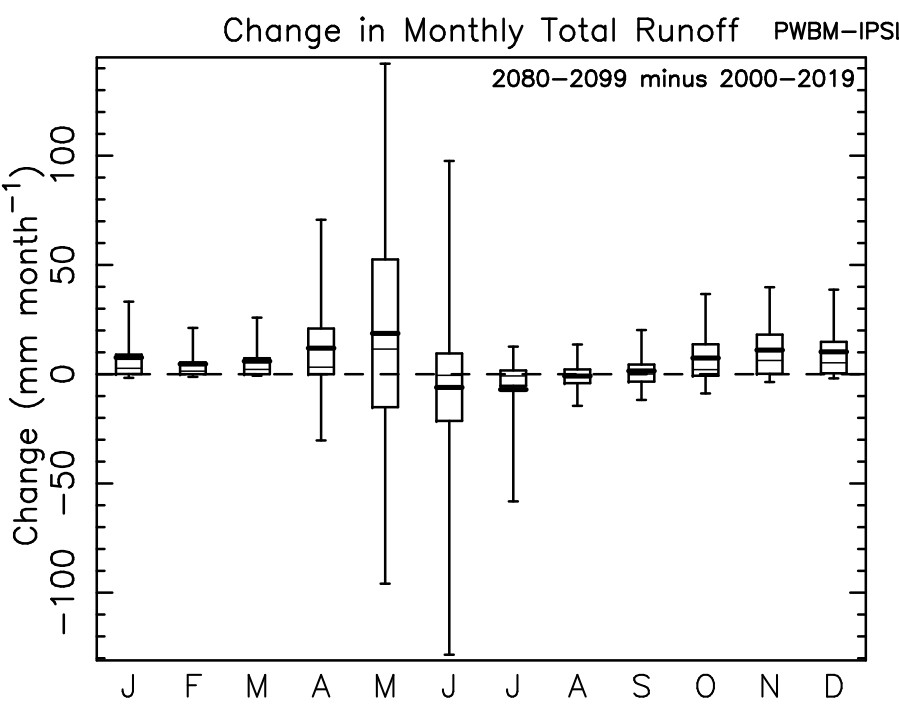

Figure S10: Distribution in change in monthly total runoff (mm month$^{-1}$) between early and late century periods for all pan-Arctic grid cells from PWBM-IPSL.