# Peer review of "Regime Shifts in Arctic Terrestrial Hydrology Manifested From Impacts of Climate Warming"

_The Cryosphere, 2023_

## Referee Comment (RC2)

[referee-annotated manuscript omitted]

---

## Author Comment (AC1)

Significant uncertainty exists about how increasing temperatures and changing precipitation patterns will affect Arctic hydrological systems and, in turn, freshwater exports and associated biogeochemical fluxes to the oceans. Arctic hydrology is characterized by strong coupling between flow and thermal processes and is generally not well represented in Earth System Models. Rawlins and Karmalkar use the process-based model PWBM and two climate projections in a strong-warming scenario to assess changes in river flows across the Arctic. The study is well-designed and carefully executed, the manuscript is clear and well written, and the results will be of interest to the readership of TC. However, I have one concern/question that needs to be addressed.

**Main question/concerns:**

In Figure 1, which addresses model confirmation/evaluation for ALT, we can see good agreement in the mean between TPDC and PWBM forced by W5E5 (Figure 1d) but comparing 1a and 1b visually, it looks like the PWBM model is predicting significantly shallower active layer over the northernmost permafrost zone and deeper active layer in the southern parts of permafrost zone. In other words, TPDC and PWBM are producing very different trends in ALT with latitude. The fact that the two produce similar mean values is not an adequate criterion for judging the reliability of the model. An image showing the spatial distribution of the differences is needed here (e.g. like 1a and b, but for differences between TPDC and PWBM). In addition, a better metric would be the something like root mean-difference or similar metric that integrates differences across the permafrost zone. An explanation for the differences and the different trend is needed. If there is independent information available that could lend further support to the model result, that would help build confidence in PWBM's ALT calculation.

The authors thank the reviewer for their time and effort spent in evaluating the manuscript for publication in The Cryosphere. Several problems are inherent when evaluating regional or pan-Arctic distributions of simulated active-layer thickness (ALT) against observed ALT obtained from sparse in situ networks. First, in situ ALT is obtained at a point location that may not be representative of the region in which it is location. Second, observed ALT networks are very sparse across the terrestrial Arctic. Ran et al. (2022) presented an evaluation of the TPDC dataset and its new high-resolution estimates of the permafrost thermal state. The authors opine that TPDC dataset is appropriate for use in this study. Figure 2 in Ran et al. illustrates the dearth in in situ ALT, particularly across the cold mountainous areas of western Siberia and over the northern Canadian archipelago. Moreover, they stated:

"The ALT represents the hydrothermal state near the ground surface with more spatiotemporal heterogeneity than the MAGT, which represents the thermal state of the relatively deeper ground. The vulnerability of the near-surface ground to external disturbances associated with the inconsistency of the ALT measurement method may be one of the reasons for the large uncertainty in the prediction of the ALT. **Of course, the uncertainty of ALT is considerable, especially in the vast area of western Siberia where the training data are sparse. The low spatial representativeness of training data may lead to an overestimation in several Siberian mountain regions and underestimation near the lower boundary of permafrost.** This highlights the importance and urgency of state."

In summary, Ran et al. clearly stated that the distribution of ALT in the TPDC dataset is constrained.

Rawlins et al. (2013) examined simulated ALT against observations along a transect through central Alaska. Results confirmed that simulated ALT was unbiased. The PWBM has a rich history of use in characterizing land surface hydrology across the northern high latitudes. No evidence has suggested that the model is biased. The authors are confident in the quality of the simulation of soil freeze -thaw, particularly given inherent challenges in modeling pan-Arctic hydrology, and the few number of studies of this type. Recent large-scale modeling studies of coupled permafrost and hydrology reveal challenges in ALT simulation. Lawrence et al. (2019) found strong connections between snow density, forcing data, and ALT. They noted large differences in simulated permafrost distribution and ALT between two different forcing data sets used, which they suggested reveals an important aspect of uncertainty in permafrost modeling. They also found that CLM4.5, with its low-density snow, exhibited ALT that was unrealistically deep ALT (>1 m deep) across nearly the entire permafrost domain. Paquin et al. (2014) noted a tendency for the Canadian Regional Climate Model (CRCM5) to overestimate ALT compared to observed values at Circumpolar Active Layer Monitoring program (CALM) sites. They noted simulated ALTs were overestimated moderately in very cold climate of the Canadian Arctic Archipelago, with larger overestimation of ALT for CALM sites located inland, mostly along the Mackenzie River and Alaska.

Lawrence, D.M., Fisher, R.A., Koven, C.D., Oleson, K.W., Swenson, S.C., Bonan, G., Collier, N., Ghimire, B., van Kampenhout, L., Kennedy, D. and Kluzek, E., 2019. The Community Land Model version 5: Description of new features, benchmarking, and impact of forcing uncertainty. Journal of Advances in Modeling Earth Systems, 11(12), pp.4245-4287.

Paquin, J.P. and Sushama, L., 2015. On the Arctic near-surface permafrost and climate sensitivities to soil and snow model formulations in climate models. Climate Dynamics, 44, pp.203-228.

Rawlins, M.A., Nicolsky, D.J., McDonald, K.C. and Romanovsky, V.E., 2013. Simulating soil freeze/thaw dynamics with an improved pan-Arctic water balance model. Journal of Advances in Modeling Earth Systems, 5(4), pp.659-675.

Regarding model evaluation statistics, metrics which rely on squared differences are known to be problematic (Willmott et al., 2005; Hodson, 2022) Indeed the RMSE in particular is RMSE is inappropriate because it is a function of 3 characteristics of a set of errors, rather than of one (the average error). RMSE varies with the variability within the distribution of error magnitudes and with the square root of the number of errors, as well as with the average-error magnitude (MAE). Interpretation problems can thus arise because sums-of-squares-based statistics do not satisfy the triangle inequality (Willmott et al., 2009). The authors feel strongly that MAE is a more natural measure of average error, and evaluations and inter-comparisons between models and observations should be based upon it.

Hodson, T.O., 2022. Root-mean-square error (RMSE) or mean absolute error (MAE): When to use them or not. Geoscientific Model Development, 15(14), pp.5481-5487.

Willmott, C.J. and Matsuura, K., 2005. Advantages of the mean absolute error (MAE) over the root mean square error (RMSE) in assessing average model performance. Climate research, 30(1), pp.79-82.

Willmott, C.J., Matsuura, K. and Robeson, S.M., 2009. Ambiguities inherent in sums-of-squares-based error statistics. Atmospheric Environment, 43(3), pp.749-752.

**Other comments:**

The manuscript has a good summary of the PWBM at about the right level of detail, but neglects one important piece of information: what is the spatial structure? I presume it's not fully 3D, but a collection of independent columns with parameterized landscape runoff and routing through a river network? A brief description would help.

The spatial domain encompassing the terrestrial pan-Arctic as defined in this study involves 35,693 grid cells of 25x25 km resolution. These are, indeed, 35,693 columns in which water and energy interact with the soil and vegetation. Implementation of a routing routine (not used in this study) would make the model 3D. Results herein are based on the interaction of soil physics and hydrology as it manifest in changes in runoff, both spatially, with depth, and with time, both a seasonal component and difference from recent past to end of century. Methods section will be augmented with additional language to provide additional detail for the interested reader.

It would be useful to know what fraction of the contributing area for the major rivers comes from nonpermafrost regions. This information would allow the reader to judge whether the results are coming mostly from trends in precipitation or from deepening of the ALT in a warming climate.

The authors propose to compute the statistics and add statements based on them. However, the close correspondence between changes in simulated net precipitation and simulated runoff suggest that net precipitation, rather than de-watering permafrost, is forcing the changes. As described in the paper, and specifically shown in figure 7, the runoff *increases* will arise mostly from colder northern areas, which tend to be underlain by permafrost. Physically, a deepening ALT would tend to store more water that could potentially be evapotranspired, and advected away, at a later time. The author feel that the results clearly illustrate that changes in net precipitation---increases in colder areas where runoff/precipitation rates tend to be high because of frozen ground, and decreases in southerly areas of unfrozen ground, are the dominant factor. Deepening ALT is playing a role in the transition to proportionally more subsurface runoff and increasing flows in autumn. The latter change is supported by many recent studies that are based on in situ observations. The addition of the contributing area statistic may allow readers to gain insights, and so will be added.

I'm not sure what is meant by "seasonally maximum ALT" in Figs 1 and 4 as ALT is already the annual maximum thaw depth. Isn't this just ALT?

The authors appreciate the question regarding seasonally maximum ALT. Actually, ALT is the thickness of the active (thawed) layer at any time. For example, during the early part of the thawed (warm) season, the active layer is typically deepening. ALT in a given area may be, for example, 10 cm in mid

June, 20 cm in mid July, and a maximum of, say, 30 cm in mid to late August. Thus, the authors, as other researcher have done, feel it is important to make clear to the reader that the metric of most relevance for validation is the maximum depth that occurs during the thawed season.

The manuscript correctly notes that subsidence, which is neglected in the model, may result in more discharge. It may also be worth noting that the cited modeling study by Painter et al. (2023) was specific to polygonal tundra so the effect on large river basins will depend on the fraction of those basins that contain polygonal tundra.

This is a valid and important point. The modeling study of Painter et al. (2023) is an important contribution to the growing body of evidence thaw permafrost thaw is impacting Arctic terrestrial hydrology. The originally submitted manuscript did make reference to polygonal tundra (line 574). Additionally mention will be added in the Introduction section.

---

## Author Comment (AC2)

In this manuscript, the authors simulate frozen ground and streamflow across a pan-Arctic domain. They simulate increased runoff, higher proportions of subsurface flow and a loss of permafrost. A notable finding is a regional shift in runoff towards more northerly locations, which have higher amounts of soil carbon. This leads to the conclusion that there could be enhanced carbon fluxes to the Arctic Ocean. The paper is well written and presented. I have several suggestions, which are embedded in the attached document, so I will only summarize in these comments.

The only major concern I have is the validation of the representation of frozen ground. It is unclear how the authors represent discontinuous permafrost within a model grid. Furthermore, I am not sure how the simulated ground thermal state is compared to observations. I struggle with using permafrost class to validate simulated active layer thickness. A better model validation approach is necessary.

The authors appreciate the reviewer's assessment and comments. There are two components to clarify with respect to the validation of frozen ground.

Regarding permafrost classification, permafrost class from the International Permafrost Association (IPA) dataset (Brown et al., 1997) was presented simply to illustrate the spatial pattern in permafrost distribution for the benefit of a broad readership. It was not used to validate simulated active-layer thickness (ALT). For that the authors leveraged a new dataset, (TPDC), which was produced using a machine-learning approach that incorporated in situ data of permafrost thermal state and ALT (Ran et al., 2022). Results of that comparison are documented in figure 1d of the submitted manuscript. While the TPDC data set is state-of-art, there are inherent uncertainties in its spatial representation of ALT, as discussed by Ran et al. (2022), and as discussed in the reply to review #1. In essence, Ran et al. specifically stated that the distribution coming out of the machine-learning method is, in their estimation, constrained. Thus the validation results shown in figure 1d are excellent given the known challenges in simulating soil temperatures.

*Brown, J., O.J. Ferrians, Jr., J.A. Heginbottom, and E.S. Melnikov, eds. 1997. Circum-Arctic map of permafrost and ground-ice conditions. Washington, DC: U.S. Geological Survey in Cooperation with the Circum-Pacific Council for Energy and Mineral Resources. Circum-Pacific Map Series CP-45, scale 1:10,000,000, 1 sheet.*

*Ran, Y., Li, X., Cheng, G., Che, J., Aalto, J., Karjalainen, O., Hjort, J., Luoto, M., Jin, H., Obu, J. and Hori, M., 2022. New high-resolution estimates of the permafrost thermal state and hydrothermal conditions over the Northern Hemisphere. Earth system science data, 14(2), pp.865-884.*

Regarding discontinuous permafrost, there is no dealing with that, per se. The spatial domain is discretized into an array of grid cells. To determine permafrost state for a grid cell, soil temperatures that vary with depth (23 layers to 60 meters) are examined. A grid cell is deemed to be permafrost or seasonally frozen (non-permafrost) based on the grid cell temperature profile. In the case where soil temperatures are well simulated, one can assume that there is discontinuous permafrost in regions where many grid cells classified as permafrost interface with many grid cells classified as seasonally frozen. Similar to other studies using a land surface model, permafrost state is a binary classification. No subgrid variability is implemented. The authors are confident that no systematic bias exists with this approach. That is, in areas where a majority of cells are classified as permafrost, a significant amount of terrain would, in the real world, contain permafrost, with no permafrost on, for example, south

facing slopes. Likewise, in areas where the model simulation points to no permafrost, much of the terrain would have no frozen ground to depth, with patchy areas of permafrost on, for example, north facing slopes. Lines 179-186 in the submitted manuscript describe how the presence of permafrost is determined. More relevant information on simulation ALT and associated challenges is articulated in response to reviewer RC1.

Could I suggest the authors present panels of differences (i.e., GLEAN subtract PWBM) in Figures 1 and 2 to show how different the model is from observations? This would reveal where uncertainty is highest. This is important because one of the main takeaways is the importance of regional differences in responses to warming and the impact this has on freshwater fluxes to the Arctic Ocean.

Further to this point, the conclusion that the model performs well is based on an assessment of the model's performance over the entire domain. This is why I am suggesting those extra panels in Figures 1 and 2 as they will help in assessment of uncertainty across sub-domains.

Finally, I like the paper and what the authors are trying to accomplish. I am hoping that addressing these suggestions will make the conclusions in the final version of the paper a bit more defensible, and impactful. On this note, I think the conclusions could be even punchier. Please see my suggestions in the marked-up version.

Thanks for the chance to review the paper.

Difference at grid cell level have been computed. In light of this comment the authors propose to create map panels, add them, and describe differences in a paragraph to be added to section 3 'Model Validation'.

Specific comments:

Lines 115-119: The authors are willing to add a statement to that effect.

Line 122: Appreciate the suggestion.

Line 142: The authors agree, and can do this.

Line 190: Language corrections noted.

Line 195: Only simulated ALT was examined from all three forcings. This in light of the importance of simulated ALT. The authors have clarified the excellent performance of the model simulation of ALT in responses to reviewer #1. The manuscript describes all the remaining results that consistently leverage just the W5E5-based estimates for model validation.

Line 216: The phrase "business as usual" is appropriate.

Line 263: The authors are willing to create the map panels and add them.

Comment on figure 1: Permafrost class addressed in major comments section above. A difference map for ALT to be added.

Line 277: Respectfully, differences of less than 13% is an excellent result for numerical model simulation of Arctic river discharge.

Line 293: Map panels of difference in figures 1 and 2 to be added.

Line 313: The authors disagree with the reviewers assertion. No modeling studies published in recent years suggest that rainfall in Arctic regions will not increase.

Line 328: There is a large body of prior research based on results using this model. The authors believe that addition of extensive model validation discussion would be awkward given the length and breadth of the submitted manuscript. The authors are willing to add map panels of differences shown in figures 1 and 2.

Line 328: The nature of grid cell permafrost classification is articulated above.

Line 353: The authors propose addition of a paragraph in the model validation section which will focus on regional uncertainties based on map graphics in figures 1 and 2.

Line 356: Yes. The close correspondence between P-ET and runoff support the statement that changes (increases and decreases) in net precipitation are the main driver of runoff changes.

Line 373: Runoff is defined at lines 157-160. The authors are willing to add another sentence.

Caption figure 6: Change to be made on revision.

Line 404: This is an artifact of the interpolation of IPSL climate fields.

Line 410: Respectfully, there is evidence, though not explicitly demonstrated with a figure graphic in the present manuscript. The authors feel that additional model validation of this nature is far beyond the scope of the present study. It would require assembly of monthly climatologies of river discharge for major rivers, translated to runoff per unit depth across contributing watersheds, and subsequent assessment against model simulations. Rather, we show here prior assessments made during an earlier study. The authors believe that the model simulations are robust enough to support the analysis and statements characterized at line 410.

[Figure]

Line 415: Will be moved to Methods on revision.

Line 421: Statement is correct; the mean change in June and July are not significant due to the broad distribution of change across the domain as shown by the boxplots. A phrase will be added to the sentence to clarify this point.

Line 465: Independent clause is based on the amount of net precipitation change in the two simulations described. De-watering not quantified in the present study. Can reword to state that de-watering of permafrost plays a much smaller role, citing McClelland et al. (2004).

*McClelland, J. W., Holmes, R. M., Peterson, B. J., and Stieglitz, M. (2004), Increasing river discharge in the Eurasian Arctic: Consideration of dams, permafrost thaw, and fires as potential agents of change, J. Geophys. Res., 109, D18102, doi:10.1029/2004JD004583.*

Line 492-494: Yes, the statement is based entirely on model simulations and examination of associated outputs. The monthly distributions were examined. Moreover, there is no increase in model simulated SWE in autumn. Figures not shown in manuscript given the large number of existing graphics.

[Figure]

Line 506: Agreed with reviewer comment. Clause regarding other studies will be added on revision.

Line 539: Increased soil thaw in the simulations in areas rich with soil carbon support the second part, and is consistent with studies based on river sampling.

Line 546: Agree that details of approximations for PET would overwhelm readers. The descriptions in this paper are more succinct given that three more detailed papers have been published in recent years. A model simulating AET is too computationally expensive for a study of this type, and, moreover, forcing data are lacking. Using a function like Penman-Monteith would introduce much more uncertainty. A cost benefit analysis suggests that using the Haman function is appropriate. After all, the

study is not focused on accuracy of land-atmosphere water fluxes. That said, mention of Hamon function will be added to Methods section.

Line 575: The authors appreciate the comment. The authors opine that uncertainties have been articulated where appropriate. The PWBM is suitably physically scaled for a study of this type. It is not a land-surface model of the type used in coupled climate model simulations. Extensive discussion of model uncertainties would be a different paper altogether. The present manuscript is not focused on a rigorous assessment of validation exercises with the PWBM. The present results build on a rich history of prior studies, each of which contain model validations suitable for the study objectives and goals. Detailed validation at regional scale is beyond the scope of the present study which seeks to gain insights into future trajectories. Moreover, all of the results, namely, that runoff will shift northward, and to more subsurface flow and toward later in autumn, are consistent with recent studies that leverage river sampling and in situ ground measurements. In revision the authors will add statements based on difference maps arising from fields shown in figures 1 and 2.

---

## Author Response (AR2)

Change made to caption of figure 1 to refer to **annual** maximum ALT.

Reviewer #1

The new Figure 4d is a good start at evaluating the model's predicted ALT. There are clearly very large differences between the TPDC dataset and the model output over large regions. For that reason, we need to see some quantification of the error. For example, what is the mean absolute error? This applies to ET and sublimation as well.

We assume reviewer #1 was referring to figure 1d, as figure 4 does not show ALT. For the benefit of readers we have added a panel to figure 1e (boxplots) to show the mean absolute errors and mean bias errors. As stated in the first paragraph of section Model Validation, the creators of the TPDC dataset have made clear to the research community that the distribution of TPDC ALTs is very likely (by their estimation) more narrow than reality. They attribute this to sampling bias. It is thus reasonable to assume that the MAEs that we report here are inflated. We also now report MAEs for sublimation and ET. Lines 304 and 306 of revised manuscript.

Reviewer #2

I only have one suggestion; which is totally optional. The language describing how subsurface runoff is conceptualized and simulated in the study could be clarified. For instance in the abstract maybe substitute "... the proportion of subsurface to total runoff ..." with " .... the proportion of total runoff exposed to subsurface pathways ...". Similarly, on Line 173 in the Modelling Approach section the authors could say "We use the term "subsurface runoff" for the water flux that has followed subsurface pathways into the stream."

I hope this is not being too pedantic.

We have made the changes in the abstract. The phrase now reads: ...**while the proportion of runoff emanating from subsurface pathways is projected…**

We've also implemented the change as suggested for the Modeling Approach section.